# ON WEIGHT-SHARING AND BILEVEL OPTIMIZATION IN ARCHITECTURE SEARCH

## ABSTRACT

Weight-sharing—the simultaneous optimization of multiple neural networks using the same parameters—has emerged as a key component of state-of-the-art neural architecture search. However, its success is poorly understood and often found to be surprising. We argue that, rather than just being an optimization trick, the weight-sharing approach is induced by the relaxation of a structured hypothesis space, and introduces new algorithmic and theoretical challenges as well as applications beyond neural architecture search. Algorithmically, we show how the geometry of ERM for weight-sharing requires greater care when designing gradient-based minimization methods and apply tools from non-convex non-Euclidean optimization to give general-purpose algorithms that adapt to the underlying structure. We further analyze the learning-theoretic behavior of the bilevel optimization solved by practical weight-sharing methods. Next, using kernel configuration and NLP feature selection as case studies, we demonstrate how weight-sharing applies to the *architecture search* generalization of NAS and effectively optimizes the resulting bilevel objective. Finally, we use our optimization analysis to develop a simple exponentiated gradient method for NAS that aligns with the underlying optimization geometry and matches state-of-the-art approaches on CIFAR-10.

## 1 INTRODUCTION

Weight-sharing neural architecture search (NAS) methods have achieved state-of-the-art performance while requiring computation training of just a single *shared-weights network* (Pham et al., 2018; Li and Talwalkar, 2019; Liu et al., 2019). However, weight-sharing remains poorly understood. In this work, we present a novel perspective on weight-sharing NAS motivated by the key observation that these methods subsume the architecture hyperparameters as another set of learned parameters of the shared-weights network, in effect extending the hypothesis class. An important ramification of this insight is that weight-sharing is not NAS-specific and can be used to tune hyperparameters corresponding to parameterized feature maps of the input data. We refer this larger subset of hyperparameter optimization problems as *architecture search*, and we study the following two questions associated with weight-sharing applied to the architecture search problem:

*1. How can we efficiently optimize the objective induced by applying weight sharing to architecture search, namely minimizing empirical risk in the joint space of model and architecture parameters?*
For large structured search spaces that preclude brute force search, a natural approach to architecture search with weight-sharing is to use gradient-based methods to minimize the empirical risk over a continuous relaxation of the discrete space (Liu et al., 2019). Although this has allowed NAS researchers to apply their preferred optimizers to determine architecture weights, it is far from clear that the success of established methods for unconstrained optimization in training neural networks will naturally extend to these constrained and often non-Euclidean environments. As we foresee that architecture search spaces will continue to become more complex and multi-faceted, we argue for and develop a more principled, *geometry-aware* formulation of the optimization problem. Drawing upon the mirror descent meta-algorithm (Beck and Teboulle, 2003) and successive convex approximation, we give non-asymptotic stationary-point convergence guarantees for the empirical risk minimization (ERM) objective associated with weight-sharing via algorithms that simultaneously connect to the underlying problem structure and handle the alternating-block nature of the architecture search. Our guarantees inform the design of gradient-based weight-sharing methods by explicitly quantifying the impact of optimizing in the right geometry on convergence rates.

*2. What are the generalization benefits of solving a bilevel optimization for the architecture search problem commonly considered in practice?* At its core, the goal of architecture search is to find a configuration that achieves good generalization performance. Consequently, a bilevel objective that optimizes the architecture weights using a separate validation loss is commonly used in practice in lieu of the ERM objective naturally induced by weight sharing (Pham et al., 2018; Liu et al., 2019; Cai et al., 2019). The learning aspects of this approach have generally been studied in settings with much stronger control over the model complexity (Kearns et al., 1997). We provide generalization guarantees for this objective over structured hypothesis spaces associated with a finite set of architectures; this leads to meaningful bounds for simple feature map selection problems as well as insightful results for the NAS problem that depend on the size of the space of global optima.

To validate our theoretical results, we conduct empirical studies of weight-sharing in two settings: (1) shallow feature map selection, i.e., tuning the hyperparameters of kernel classification and NLP featurization pipelines, and (2) CNN neural architecture search. In (1) we demonstrate that weight-sharing efficiently optimizes the bilevel objective and achieves low generalization error with respect to the best architecture setting. For (2), motivated by insights from our convergence analysis, we develop a simple exponentiated gradient version of DARTS (Liu et al., 2019) called EDARTS that better exploits the geometry of the optimization problem. We evaluate EDARTS on the design of CNN architectures for CIFAR-10 and demonstrate that EDARTS finds better architectures than DARTS in less than half the time. We also achieve very competitive results relative to state-of-the-art architectures when using an extended evaluation routine.

**Related Work:** Our work on optimization for weight-sharing benefits from the literature on first-order stochastic optimization (Hazan and Kale, 2014; Beck, 2017) and in particular the mirror descent framework (Beck and Teboulle, 2003). Specifically, we use successive convex approximation (Razaviyayn et al., 2013; Mairal, 2015) to show convergence of alternating minimization and derive geometry-dependent rates comparable to existing work on non-convex stochastic mirror descent (Dang and Lan, 2015; Zhang and He, 2018). Our result generalizes to the constrained, non-Euclidean, and multi-block setting an approach of Agarwal et al. (2019) for obtaining non-convex convergence from strongly convex minimization, which may be of independent interest. Previous optimization results for NAS have generally only shown bounds on auxiliary quantities such as regret that are not well-connected to the learning objective (Noy et al., 2019; Nayman et al., 2019; Carlucci et al., 2019) or have only given monotonic improvement or asymptotic guarantees (Akimoto et al., 2019; Yao et al., 2019). However, due to the generality of mirror descent, the approaches in the middle three papers can be seen as special cases of our analysis. Finally, our analysis of the properties of the bilevel optimization is related to work on model selection (Vuong, 1989; Kearns et al., 1997), but does not consider the configuration parameters as explicit controls on the model complexity. Our learning results are broadly related to hyperparameter optimization, although most work focuses on algorithmic and not statistical questions (Li et al., 2018; Kandasamy et al., 2017).

## 2 THE WEIGHT-SHARING LEARNING PROBLEM

In this section, we formalize the weight-sharing learning problem, relate it to traditional ERM, and provide examples for the case of NAS and feature map selection that we use for the rest of the paper. Our main observation is that weight-sharing for architecture search extends the hypothesis space to be further parameterized by a finite set of configurations $\mathcal{C}$. Formally, we have a structured hypothesis space $H(\mathcal{C}, \mathcal{W}) = \{h_w^{(c)} : X \mapsto Y : w \in \mathcal{W}, c \in \mathcal{C}\}$ over input space $Z = X \times Y$ with induced hypothesis subclasses $H_c = \{h_w^{(c)} : X \mapsto Y : w \in \mathcal{W}\}$ parameterized by weights from $\mathcal{W}$. The goal of learning is the usual one—find $h_w^{(c)} \in H(\mathcal{C}, \mathcal{W})$ with low population error $\ell_{\mathcal{D}}(h_w^{(c)}) = \mathbb{E}_{(x,y) \sim \mathcal{D}} \ell(h_w^{(c)}(x), y)$ for loss $\ell : Y \times Y \mapsto \mathbb{R}$. Hence, we can apply ERM as usual, with optional regularization, to select a hypothesis from the extended hypothesis space; in fact this is done by some NAS methods (e.g., Xie et al., 2019). The learning algorithm is then

$$\min_{w \in \mathcal{W}, c \in \mathcal{C}} \quad \mathcal{L}_T(h_w^{(c)}) = \min_{w \in \mathcal{W}, c \in \mathcal{C}} \quad \frac{1}{|T|} \sum_{(x,y) \in T} \ell(h_w^{(c)}(x), y) + R_{\mathcal{W}}(w) + R_{\mathcal{C}}(c) \quad (1)$$

for block specific regularizers $R_{\mathcal{W}}$ and $R_{\mathcal{C}}$. Note that in the absence of weight-sharing, we would need to learn a separate hypothesis $h_{w_c}^{(c)}$ for each hypothesis subclass $H_c$. Although a brute force approach to selecting a hypothesis from $H(\mathcal{C}, \mathcal{W})$ via ERM would in effect require this as well,

our subsequent examples demonstrate how the weight-sharing construct allows us to apply more efficient gradient-based optimization approaches, which we study in Section 3.

**Feature Map Selection:** In this setting, the structure is induced by a set of feature transformations $\mathcal{C} = \{\phi_i : X \mapsto \mathbb{R}^n \text{ for } i = 1, \dots, k\}$, so the hypothesis space is $\{f_w(\phi_i(\cdot)) : w \in \mathcal{W}, \phi_i \in \mathcal{C}\}$ for some $\mathcal{W} \subset \mathbb{R}^d$. Examples of feature map selection problems include tuning kernel hyperparameters for kernel ridge classification and tuning NLP featurization pipelines for text classification. In these cases $f_w$ is a linear mapping $f_w(\cdot) = \langle w, \cdot \rangle$ and $\mathcal{W} \subset \mathbb{R}^n$.

**Neural Architecture Search:** Weight-sharing methods almost exclusively use micro cell-based search spaces for their tractability and additional structure (Pham et al., 2018; Liu et al., 2019). These search spaces can be represented as directed acyclic graphs (DAGs) with a set of ordered nodes $N$ and edges $E$. Each node $x^{(i)}$ in the DAG is a feature representation and each edge $o^{(i,j)}$ is an operation on the feature of node $j$ passed to node $i$ and aggregated with other inputs to form $x^{(j)}$, with the restriction that a given node $j$ can only receive edges from prior nodes as input. Hence, the feature at a given node $i$ is $x^{(i)} = \sum_{j<i} o^{(i,j)}(x^{(j)})$. Search spaces are then specified by the number of nodes, the number of edges per node, and the set of operations $O$ that can be applied at each edge.

In this case the structure $\mathcal{C} \subset \{0,1\}^{|E||O|}$ of the hypothesis space is the set of all valid architectures for this DAG encoded by edge and operation decisions. Treating both weights $w \in \mathcal{W}$ and architecture decision $c \in \mathcal{C}$ as parameters, weight-sharing methods train a single shared-weights network $h_w^{(c)} : X \mapsto Y$ encompassing all possible functions within the search space. Therefore, the shared-weights network includes all possible edges between nodes and all possible operations per edges. In addition to the *weights* $w \in \mathcal{W}$ corresponding to all the operations, the shared-weights network also takes *architecture weights* $c \in \mathcal{C}$ as input, where $c_o^{(i,j)}$ indicates the weight given to operation $o$ on edge $(i,j)$ so that the feature of a given node $i$ is $x^{(i)} = \sum_{j<i} \sum_{o \in O} c_o^{(i,j)} o^{(i,j)}(x^{(j)})$.

Gradient-based weight-sharing methods apply continuous relaxations to the architecture search space in order to compute gradients. Some methods like DARTS (Liu et al., 2019) and its variants (Chen et al., 2019; Laube and Zell, 2019; Hundt et al., 2019; Liang et al., 2019; Noy et al., 2019; Nayman et al., 2019) relax the search space by considering a mixture of operations per edge and then discretize to a valid architecture in the search space. With the mixture relaxation, we replace all $c \in \{0,1\}^{|E||O|}$ in the above expressions by continuous counterparts $\theta \in [0,1]^{|E||O|}$, with the constraint that $\sum_{o \in O} \theta_o^{(i,j)} = 1$, i.e., the architecture weights for operations on each edge sum to 1. Other methods like SNAS (Xie et al., 2019), ASNG-NAS (Akimoto et al., 2019), and ProxylessNAS (Cai et al., 2019) assume a parameterized distribution $p_\theta$ from which architectures are sampled. By substituting continuous parameters $\theta \in \Theta$ in for discrete parameters $c \in \mathcal{C}$, we are able to use gradient-based methods to optimize (1). We address the question of how to effectively use gradient optimization for weight-sharing in the next section.

## 3   GEOMETRY-AWARE OPTIMIZATION WITH WEIGHT-SHARING

While continuous relaxation enables state-of-the-art results, architecture search remains expensive and noisy, with state-of-the-art mixture methods requiring second-order computations (Liu et al., 2019) and probabilistic methods suffering from high variance policy gradients (Xie et al., 2019). Moreover, while the use of SGD to optimize network weights is a well-tested approach, architecture weights typically lie in constrained, non-Euclidean geometries in which other algorithms may be more appropriate. Recognizing this, several efforts have attempted to derive a better optimization schemes; however, the associated guarantees for most of them hold for auxiliary objectives, such as regret of local configuration decisions, that are not connected to the optimization objective (1) and ignore the two-block nature of the problem (Noy et al., 2019; Nayman et al., 2019; Carlucci et al., 2019). While Akimoto et al. (2019) do consider an alternating descent method for the training objective, their results are asymptotic and certainly do not indicate any finite-time convergence.

In this section we address these difficulties by showing that the mirror descent (MD) framework (Beck and Teboulle, 2003) is the right tool for designing algorithms in the block optimization problems that occur in architecture search. We describe how such geometry-aware gradient algorithms lead to faster stationary-point convergence; as we will show in Section 5, this yields simple, principled, and effective algorithms for large-scale multi-geometry problems such as NAS.

---

**Algorithm 1:** Two geometry-aware optimization algorithms for multi-block optimization for a $\beta$-strongly-smooth function over $\mathcal{X} = \bigtimes_{i=1}^{b} \mathcal{X}_i$ given associated DGFs $\omega_i : \mathcal{X}_i \mapsto \mathbb{R}$.

---

**Input:** initialization $x^{(1)} \in \mathcal{X}$, batch-size $N \geq 1$ (for SBMD), tolerance $\varepsilon > 0$ (for ASCA)

**for** iteration $t \in [T]$ **do**

> sample $\quad i \sim \mathrm{Unif}[b] \quad$ and set $\quad x_{-i}^{(t+1)} \leftarrow x_{-i}^{(t)}$
>
> **if** *SBMD (Dang and Lan, 2015)* **then**
>> $x_i^{(t+1)} \leftarrow \arg\min_{u \in \mathcal{X}_i} \quad \langle \bar{g}, u \rangle + \beta D_{\omega_i}(u || x_i^{(t)}) \quad$ for $\quad \bar{g} = \frac{1}{N} \sum_{j=1}^{N} g_i(x^{(t)}, \zeta_{tj})$
>
> **if** *ASCA (this work)* **then**
>> $x_i^{(t+1)} \leftarrow \hat{x} \quad$ s.t. $\quad \mathbb{E}\hat{f}(\hat{x}) - \min_{u \in \mathcal{X}_i} \hat{f}(u) \leq \varepsilon \quad$ for $\quad \hat{f}(\cdot) = f(\cdot, x_{-i}^{(t)}) + 2\beta D_{\omega_i}(\cdot || x_i^{(t)})$

**Output:** $x \sim \mathrm{Unif}\{x^{(t)}\}_{t=1}^{T}$.

---

**Problem Geometry:** We relax optimization problems of the form (1) to the problem of minimizing a function $f : \mathcal{X} \mapsto \mathbb{R}$ over a convex product space $\mathcal{X} = \bigtimes_{i=1}^{b} \mathcal{X}_i$ consisting of blocks $i$, each with an associated norm $\|\cdot\|_{(i)}$. For example, in typical NAS we can set $\mathcal{X}_1$ to be the set-product of $|E|$ simplices over the operations $O$ and the associated norm to be $\|\cdot\|_1$ while $\mathcal{X}_2 = \mathcal{W} \subset \mathbb{R}^d$ is the space of network weights and the associated norm is $\|\cdot\|_2$. To each block $i$ we further associate a *distance-generating function* (DGF) $\omega_i : \mathcal{X} \mapsto \mathbb{R}$ that is 1-strongly-convex w.r.t. $\|\cdot\|_{(i)}$; each is associated to a *Bregman divergence* $D_{\omega_i}(x || y) = \omega_i(x) - \omega_i(y) - \langle \nabla \omega_i(y), x - y \rangle$, a standard measure of distance (Bregman, 1967). For example, in the Euclidean case using $\omega_2(\cdot) = \frac{1}{2} \| \cdot \|_2^2$ yields the usual squared Euclidean distance; over the probability simplex we often use the entropy $\omega_1(\cdot) = \langle \cdot, \log(\cdot) \rangle$, which is 1-strongly-convex w.r.t. $\|\cdot\|_1$ and for which $D_{\omega_1}$ is the KL-divergence.

Given an unbiased gradient estimate $g(x_t, \zeta) = \mathbb{E}_\zeta \nabla f(x_t)$, the (single-block) stochastic MD step is

$$x_{t+1} = \arg\min_{u \in \mathcal{X}} \quad \eta \langle g(x_t, \zeta), u \rangle + D_\omega(u || x_t) \tag{2}$$

for some learning rate $\eta > 0$. In the Euclidean setting this reduces to SGD, while with the entropic regularizer the iteration becomes equivalent to *exponentiated gradient*. Key to the guarantees for MD is the fact that the dual norm of the problem geometry is used to measure the second moment of the gradient; if every coordinate of $g$ is bounded a.s. by $\sigma$, then under Euclidean geometry the dependence is on $\mathbb{E}\|g(x)\|_{2,*}^2 = \mathbb{E}\|g(x)\|_2^2 \leq \sigma^2 d$ while using entropic regularization it is on $\mathbb{E}\|g(x)\|_{1,*}^2 = \mathbb{E}\|g(x)\|_\infty^2 \leq \sigma^2$. Thus in such constrained $\ell_1$ geometries mirror descent can yield dimension-free convergence guarantees. While in this paper we focus on the benefit in this simple case, the MD meta-algorithm can be used for many other geometries of interest in architecture search, such as for optimization over positive-definite matrices (Tsuda et al., 2005).

**Algorithms and Guarantees:** We propose two methods for the above multi-block optimization problems: stochastic block mirror descent (SBMD) and alternating successive convex approximation (ASCA). At each step, both schemes pick a random coordinate $i$ to update; SBMD then performs a mirror descent update similar to (2) but with a batched gradient, while ASCA optimizes a strongly-convex surrogate function using a user-specified solver. Note that both methods require that $f$ is $\beta$-strongly-smooth w.r.t. each block's norm $\|\cdot\|_{(i)}$ to achieve convergence guarantees, a standard assumption in stochastic non-convex optimization (Dang and Lan, 2015; Agarwal et al., 2019). This condition holds for the architecture search under certain limitations, such as a restriction to smooth activations. In the supplement we also show that in the single-block case ASCA converges under the more general relative-weak-convexity criterion (Zhang and He, 2018).

We first discuss SBMD, for which non-convex convergence guarantees were shown by Dang and Lan (2015); this algorithm is the one we implement in Section 5 for NAS. A first issue is how to measure stationarity in constrained, non-Euclidean geometries. In the single-block setting we can set a smoothness-dependent constant $\lambda > 0$ and measure how far the *proximal gradient operator* $\mathrm{prox}\nabla_\lambda(x) = \arg\min_{u \in \mathcal{X}} \lambda \langle \nabla f(x), u \rangle + D_\omega(u || x)$ is from a fixed point, which yields the *projected gradient measure* $\|\mathcal{G}_\lambda(x)\|^2 = \frac{x - \mathrm{prox}\nabla_\lambda(x)}{\lambda}$. Notably, in the unconstrained Euclidean case this yields the standard stationarity measure $\|\nabla f(x)\|_2^2$. For the multi-block case we replace $D_\omega(u || x) = \sum_{i=1}^{b} D_\omega(u_i || x_i)$ and use the pseudo-norm $\|x\|^2 = \sum_{i=1}^{b} \|x_i\|_{(i)}^2$. Then $x \in \mathcal{X}$ is

an $\varepsilon$-*stationary point* of $f$ w.r.t. the projected gradient if $\|\mathcal{G}_\lambda(x)\| \leq \varepsilon$. In the non-Euclidean case the norm of the projected gradient measures how far we are from satisfying a first-order optimality condition, namely how far the negative gradient is from being in the normal cone of $f$ to the set $\mathcal{X}$ (Dang and Lan, 2015, Proposition 4.1). For this condition, Algorithm 1 has the following guarantee:

**Theorem 3.1** (Dang and Lan (2015)). *If $f$ is $\beta$-strongly-smooth, $F = f(x^{(1)}) - \min_{u \in \mathcal{X}} f(u)$, and* $\mathbb{E}_\zeta \|g_i(x,\zeta)\|^2_{(i),*} \leq G_i^2 \ \forall \ i \in [b]$ *then SBMD needs* $\mathcal{O}\left(\frac{\beta F b^2}{\varepsilon^4} \sum_{i=1}^b G_i^2\right)$ *oracle calls to reach an* $\varepsilon$-*stationary-point $x \in \mathcal{X}$ as measured by $\|\mathcal{G}_{\frac{1}{\beta}}(x)\|$.*

We next provide a guarantee for ASCA in the form of a reduction to strongly-convex optimization algorithms. This is accomplished by the construction of the solving of a surrogate function at each iteration, which in the Euclidean case is effectively adding weight-decay. ASCA is useful in the case when efficient strongly-convex solvers are available for some or all of the blocks; for example, this is frequently the case for feature map selection problems such as our kernel approximation examples, which employ $\ell_2$-regularized ERM in the inner optimization. Taking inspiration from Zhang and He (2018), for ASCA we analyze a stronger notion of stationarity that upper-bounds $\|\mathcal{G}_\lambda(x)\|^2$, which we term the *projected stationarity measure*: $\Delta_\lambda(x) = \frac{D_\omega(x\|x^+) + D_\omega(x^+\|x)}{\lambda^2}$ for $x^+ = \text{prox}\nabla_\lambda(x)$. Note that this is *not* the same measure used by Zhang and He (2018), although in the appendix we show that in the single-block case our result holds for their notion also.

**Theorem 3.2.** *If $f$ is $\beta$-strongly-smooth and $F = f(x^{(1)}) - \min_{u \in \mathcal{X}} f(u)$ then after $T$ iterations ASCA yields a point $x \in \mathcal{X}$ s.t. $\mathbb{E}\|\mathcal{G}_{\frac{1}{4\beta}}(x)\|^2 \leq \mathbb{E}\Delta_{\frac{1}{4\beta}}(x) \leq 8\beta b\left(\frac{F}{T} + \varepsilon\right)$, where $\varepsilon > 0$ is the solver tolerance and the expectation is over the randomness of the algorithm and associated oracles.*

*Proof Summary.* The proof generalizes a result of Agarwal et al. (2019) and is in Appendix A. $\square$

Thus if we have solvers that return approximate optima of strongly-convex functions on each block then we can converge to a stationary point of the original function. The convergence rate will depend on the solver used; for concreteness we give a specification for stochastic mirror descent.

**Corollary 3.1.** *Under the conditions of Theorem 3.1, if on $i$ ASCA uses the Epoch-GD method of Hazan and Kale (2014) with divergence $D_{\omega_i}$ then $\mathcal{O}\left(\frac{\beta F b^2}{\varepsilon^4} \sum_{i=1}^b G_i^2 + \beta^2 L_{\omega_i}^2\right)$ oracle calls suffice to reach an $\varepsilon$-stationary-point $x \in \mathcal{X}$ as measured by $\mathbb{E}\sqrt{\Delta_{\frac{1}{4\beta}}(x)} \geq \mathbb{E}\|\mathcal{G}_{\frac{1}{4\beta}}(x)\|$, where $L_{\omega_i}$ is the Lipschitz constant of $\omega_i$ w.r.t. $\|\cdot\|_i$.*

This oracle complexity matches that of Theorem 3.1 apart from the extra $\beta^2 L_{\omega_i}^2$ term due to the surrogate function, which we show below is in practice a minor term that does not obviate the benefit of geometry-awareness. On the other hand, ASCA is much more general, allowing for many different algorithms on individual blocks; for example, many popular neural optimizers such as Adam (Kingma and Ba, 2015) have variants with strongly-convex guarantees (Wang et al., 2019).

**The Benefit of Geometry-Aware Optimization:** We conclude this section with a formalization of how convergence of gradient-based architecture-search algorithms can benefit from this optimization strategy. Recalling from our example, we have the architecture parameter space $\mathcal{X}_1$ consisting of $|E|$ simplices over $|O|$ variables equipped with the 1-norm and the shared weight space $\mathcal{X}_2 = \mathcal{W}$ equipped with the Euclidean norm. We suppose that the stochastic gradients along each block $i$ have coordinates bounded a.s. by $\sigma_i > 0$. Then if we run SBMD using SGD ($\ell_2$) to optimize the shared weights and exponentiated gradient ($\ell_1$) to update the architecture parameters, Theorem 3.1 implies that we reach an $\varepsilon$-stationary point in $\mathcal{O}\left(\frac{\sigma_1^2 + \sigma_2^2 d}{\varepsilon^4}\right)$ stochastic gradient computations. The main benefit here is that the first term in the numerator is not $\sigma_1^2 |E||O|$, which would be the case if we used SGD; this improvement is critical as the noise $\sigma_1^2$ of the architecture gradient can be very high, especially if a policy gradient is used to estimate probabilistic derivatives.

In the case of ASCA, we can get similar guarantees assuming the probabilities for each operation are lower-bounded by some small $\delta > 0$ and that the space of shared weights is bounded by $B$; then the guarantee will be as above except with an additional $\mathcal{O}(\log\frac{1}{\delta} + B^2)$ term (independent of $\sigma_1, \sigma_2$). While for both SBMD and ASCA the $\sigma_2^2 d$ term from training the architecture remains, this will be incurred even in single-architecture training using SGD. Furthermore, in the case of ASCA it may be improved using adaptive algorithms (Agarwal et al., 2019; Wang et al., 2019).

## 4 GENERALIZATION PROPERTIES OF THE BILEVEL FORMULATION

In Section 2, we described the weight-sharing hypothesis class $H(\mathcal{C}, \mathcal{W})$ as a set of functions non-disjointly partitioned by a set of configurations $\mathcal{C}$ sharing weights in $\mathcal{W}$ and posed the ERM problem associated with selecting a hypothesis from $H(\mathcal{C}, \mathcal{W})$. However, as mentioned in Section 1, the objective solved in practice is a bilevel problem where a separate validation set is used for architecture parameter updates. Formally, the bilevel optimization problem considered is

$$\min_{w \in \mathcal{W}, c \in \mathcal{C}} \quad \ell_V(h_w^{(c)}) \qquad \text{s.t.} \qquad w \in \arg\min_{w' \in \mathcal{W}} \quad \mathcal{L}_T(w', c) \qquad (3)$$

where $T, V \subset Z$ is a pair of training/validation sets sampled i.i.d. from $\mathcal{D}$, the upper objective $\ell_V(h_w^{(c)}) = \frac{1}{|V|} \sum_{(x,y) \in V} \ell(h_w^{(c)}(x), y)$ is the empirical risk over $V$, and $\mathcal{L}_T(w, c)$ is some objective induced by $T$. We intentionally differentiate the two losses since training is often regularized.

This setup is closely related to the well-studied problems of *model selection* and *cross-validation*. However, a key difference is that the choice of configuration $c \in \mathcal{C}$ does not necessarily provide any control over the complexity of the hypothesis space; for example, in NAS as it is often unclear how the hypothesis space changes due to the change in one decision. By contrast, the theory of model selection is often directly concerned with control of model complexity. Indeed, in possibly the most common setting the hypothesis classes are nested according to some total order of increasing complexity, forming a structure (Vapnik, 1982; Kearns et al., 1997). This is for example the case in most norm-based regularization schemes. Even in the non-nested case, there is often an explicit tradeoff between parsimony and accuracy (Vuong, 1989).

With the configuration parameters in architecture search behaving more like regular model parameters rather than as controls on the model complexity, it becomes reasonable to wonder why most NAS practitioners have used the bilevel formulation. Does the training-validation split exploit the partitioning of the hypothesis space $H(\mathcal{C}, \mathcal{W})$ induced by the configurations $\mathcal{C}$? To see when this might be true, we first note that a key aspect of the optima of the bilevel weight-sharing problem is the restriction on the model weights - that they must be in the set $\arg\min_{w \in \mathcal{W}} \mathcal{L}_T(h_w^{(c)})$ of the inner objective $\mathcal{L}_T$. As we will see, under certain assumptions this can reduce the complexity of the hypothesis space without harming performance.

First, for any sample $T \subset Z$ let $H_c(T) = \{h_w^{(c)} : w \in \arg\min_{w \in \mathcal{W}} \mathcal{L}_T(h_w^{(c)})\}$ be the *version space* (Kearns et al., 1997, Equation 6) induced by some configuration $c$ and the objective function. Second, let $N(F, \varepsilon)$ be the $L^\infty$-*covering-number* of a set of functions $F$ at scale $\varepsilon > 0$, i.e. the number of $L^\infty$ balls required to construct an $\varepsilon$-cover of $F$ (Mohri et al., 2012, Equation 3.60). These two quantities let us define a complexity measure over the shared weight hypothesis space:

**Definition 4.1.** *The **version entropy** of $H(\mathcal{C}, \mathcal{W})$ at scale $\varepsilon > 0$ induced by the objective $\mathcal{L}_T$ is* $\Lambda(H, \varepsilon, T) = \log N \left( \bigcup_{c \in \mathcal{C}} H_c(T), \varepsilon \right)$.

The version entropy is a data-dependent quantification of how much the hypothesis class is restricted by the inner optimization. For finite $\mathcal{C}$, a naive bound shows that $\Lambda(H, \varepsilon, T)$ is bounded by $\log |C| + \max_{c \in \mathcal{C}} \log N(H_c(T), \varepsilon)$, so that the second term measures the worst-case complexity of the global minimizers of $\mathcal{L}_T$. In the feature selection problem, $\mathcal{L}_T$ is usually a strongly-convex loss due to regularization and so all version spaces are singleton sets, making the version entropy $\log |\mathcal{C}|$. In the other extreme case of nested model selection the version entropy reduces to the complexity of the version space of the largest model and so may not be informative. However, in practical problems such as NAS an inductive bias is often imposed via constraints on the number of input edges.

To bound the excess risk in terms of the version entropy, we first discuss an important assumption that describes cases when we expect the shared weights approach to perform well:

**Assumption 4.1.** *There exists a good $c^* \in \mathcal{C}$, i.e. one satisfying $(w^*, c^*) \in \arg\min_{\mathcal{W} \times \mathcal{C}} \ell_\mathcal{D}(h_w^{(c)})$ for some $w^* \in \mathcal{W}$, such that w.h.p. over the drawing of training set $T \sim \mathcal{D}^{m_T}$ at least one of the minima of the optimization induced by $c^*$ and $T$ has low excess risk, i.e. w.p. $1 - \delta$ there exists $w \in \arg\min_{w' \in \mathcal{W}} \mathcal{L}_T(w', c^*)$ s.t. $\ell_\mathcal{D}(h_w^{(c^*)}) - \ell_\mathcal{D}(h_{w^*}^{(c^*)}) \leq \varepsilon_{exc}^*(m_T, \delta)$ for for excess risk $\varepsilon_{exc}^*$.*

This assumption requires that w.h.p. the inner optimization objective does not exclude all low-risk classifiers for the optimal configuration. Note that it asks nothing of either the other configurations in $\mathcal{C}$, which may be arbitrarily bad, nor of the hypotheses found by the procedure. It does however

prevent the case where one knows the optimal configuration but minimizing the provided objective $\mathcal{L}_T$ does not provide a set of good weights. Note that if the inner optimization is simply ERM over the training set $T$, i.e. $\mathcal{L}_T = \ell_T$, then standard learning-theoretic guarantees will give $\varepsilon^*_{exc}(m_T, \delta)$ decreasing in $m_T$ and increasing at most poly-logarithmically in $\frac{1}{\delta}$. With this assumption, we can show the following guarantee on solutions to the bilevel optimization.

**Theorem 4.1.** *Let $\hat{h}$ be a hypothesis corresponding to the solution of the bilevel optimization* (3). *Then under Assumption 4.1 if $\ell$ is B-bounded we have w.p. $1 - 3\delta$ that*

$$\ell_{\mathcal{D}}(\hat{h}) \leq \min_{h \in H} \ell_{\mathcal{D}}(h) + \varepsilon^*_{exc}(m_T, \delta) + \inf_{\varepsilon > 0} 3\varepsilon + \frac{3B}{2}\sqrt{\frac{2}{m_V}\left(\Lambda(H, \varepsilon, T) + \log\frac{1}{\delta}\right)}$$

*Proof Sketch.* We decompose the excess risk $\ell_{\mathcal{D}}(\hat{h}) - \ell_D(h^{c^*}_{w^*})$ as

$$\ell_{\mathcal{D}}(\hat{h}) - \ell_V(\hat{h}) \;+\; \ell_V(\hat{h}) - \ell_V(h^{(c^*)}_w) \;+\; \ell_V(h^{(c^*)}_w) - \ell_{\mathcal{D}}(h^{(c^*)}_w) \;+\; \ell_{\mathcal{D}}(h^{(c^*)}_w) - \ell_{\mathcal{D}}(h^{c^*}_{w^*})$$

The first difference is bounded by the version entropy usng the constraint on $\hat{h} \in H_c$, the second by optimality of $\hat{h}$ on $V$, the third by Hoeffding's inequality, and the last by Assumption 4.1. $\qquad\square$

As shown in the applications below, the significance of this theorem is that a bound on the version entropy guarantees excess risk almost as good as that of the (unknown) optimal configuration without assuming anything about the complexity or behavior of sub-optimal configurations.

**Feature Map and Kernel Selection:** In the feature map selection problem introduced in Section 2, $\mathcal{C} = \{\phi_i : X \mapsto \mathbb{R}^n \text{ for } i \in [k]\}$ is a set of feature maps and the inner problem $\mathcal{L}_T$ is $\ell_2$-regularized ERM for linear classification over the resulting feature vectors. The bilevel problem is then

$$\min_{c \in \mathcal{C}} \quad \ell_V(h^{(c)}_w) \qquad \text{s.t.} \qquad w = \arg\min_{w' \in \mathcal{W}} \quad \lambda\|w'\|^2_2 + \sum_{(x,y) \in T} \ell(\langle w', \phi_i(x)\rangle, y) \qquad (4)$$

Due to strong-convexity of $\mathcal{L}_T$, each map $\phi_i$ induces a unique minimizing weight $w \in \mathcal{W}$ and thus a singleton version space, therefore upper bounding the version entropy by $\log |C| = \log N$. Furthermore, for Lipschitz losses and appropriate choice of regularization coefficient, standard results for $\ell_2$-regularized ERM for linear classification (e.g. Sridharan et al. (2008)) show that Assumption 4.1 is satisfied with $\varepsilon^*_{exc}(m_T, \delta) = O\left(\sqrt{\frac{\|w^*\|^2_2 + 1}{m_T}\log\frac{1}{\delta}}\right)$. Then applying Theorem 4.1 yields

**Corollary 4.1.** *For feature map selection the bilevel optimization* (4) *with $\lambda = \sqrt{\frac{1}{m_T}\log\frac{1}{\delta}}$ yields*

$$\ell_{\mathcal{D}}(\hat{h}) \leq \min_{i \in [N]} \min_{w \in \mathcal{W}} \ell_{\mathcal{D}}(h^{(\phi_i)}_w) + \mathcal{O}\left(\sqrt{\frac{\|w^*\|^2_2 + 1}{m_T}\log\frac{1}{\delta}} + \sqrt{\frac{1}{m_V}\log\frac{N}{\delta}}\right).$$

In the special case of kernel selection using random Fourier approximation, we can apply associated generalization guarantees (Rudi and Rosasco, 2017, Theorem 1) to show that we can compete with the optimal RKHS from among those associated with one of the configurations :

**Corollary 4.2.** *In feature map selection suppose each map $\phi \in \mathcal{C}$ is associated with a random Fourier feature approximation of a continuous shift-invariant kernel that approximates an RKHS $\mathcal{H}_\phi$ and $\ell$ is the square loss. If the number of features $d = \mathcal{O}(\sqrt{m_T}\log m_T/\delta)$ and $\lambda = 1/\sqrt{m_T}$ then w.p. $1 - \delta$ we have $\ell_{\mathcal{D}}(\hat{h}) \leq \min_{\phi \in \mathcal{C}} \min_{h \in \mathcal{H}_\phi} \ell_{\mathcal{D}}(h) + \mathcal{O}\left(\frac{\log^2\frac{1}{\delta}}{\sqrt{m_T}} + \sqrt{\frac{1}{m_V}\log\frac{N}{\delta}}\right).$*

In both cases we are able to get risk bounds almost identical to the excess risk achievable if we knew the optimal configuration beforehand, up to an additional capacity term depending weakly on the number of configurations. This would not be possible with solving the regular ERM objective instead of the bilevel optimization as we would then have to contend with the possibly high complexity of the hypothesis space induced by the worst configuration.

**Neural Architecture Search:** In the case of NAS we do not have a bound on the version entropy, which now depends on all of $\mathcal{C}$. Whether the version space, and thus the complexity, of deep networks is small compared to the number of samples is unclear, although we gather some evidence. The question amounts to how many (functional) global optima are induced by a training set of size $m_T$. In an idealized spin-glass model, Choromanska et al. (2015, Theorem 11.1) suggest that the

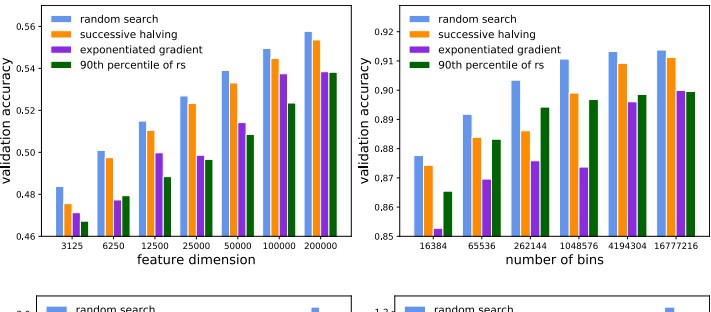

Figure 1: Validation accuracy on CIFAR-10 (left) and IMDB (right) of feature map selection with weight-sharing compared to a full sweep of random configurations. Average over 16 seeds.

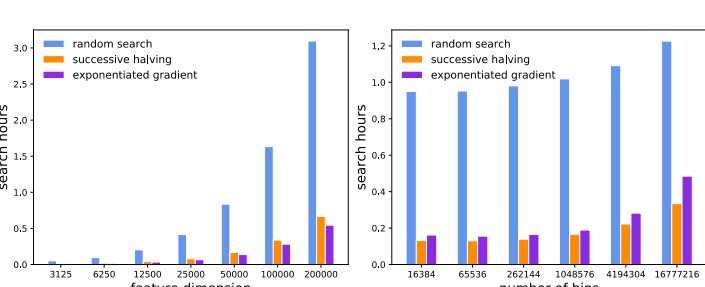

Figure 2: Search time on CIFAR-10 (left) and IMDB (right) for feature map selection. Weight-sharing finds a configuration with almost the same validation accuracy much faster than random search.

number of critical points is exponential *only* in the number of layers, which would yield a small version entropy. It is conceivable that the quantity may be further bounded by the complexity of solutions explored by the algorithm when optimizing $\mathcal{L}_T$ (Nagarajan and Kolter, 2017; Bartlett et al., 2017); indeed, we find that shared-weight optimization leads to models with smaller $\ell_2$-norm and distance from initialization than from-scratch SGD on a single network (see Appendix D.4). On the other hand, Nagarajan and Kolter (2019) argue, with evidence in restricted settings, that even the most stringent implicit regularization cannot lead to a non-vacuous uniform convergence bound; if true more generally this would imply that the NAS version entropy is quite large.

## 5 EMPIRICAL RESULTS

Here we demonstrate how weight-sharing can be used as a tool to speed up general architecture search problems by applying it to two feature map selection problems. We then validate our optimization analysis with a geometry-aware weight-sharing method to design CNN cells for CIFAR-10.

**Feature Map Selection:** Recall that here our configuration space has $k$ feature maps $\phi_i : X \mapsto \mathbb{R}^n$ with outputs passed to a linear classifier $w \in \mathbb{R}^n$, which will be the shared weights. We will approximate the bilevel optimization (3) with the inner minimization over $\ell_2$-regularized ERM $\lambda \|w\|_2^2 + \sum_{(x,y) \in T} \ell(\langle w, \phi_i(x) \rangle, y)$. Our weight-sharing procedure starts with a vector $\theta^{(1)} \in \Delta_N$ encoding a probability distribution $p_{\theta^{(1)}}$ over $[N]$ and proceeds as follows:

1. For each point $(x,y) \in T$ pick $i \sim p_{\theta^{(t)}}$ and construct featurization $f_x = \phi_i(x)$.
2. Compute shared weights $w^{(t)} = \arg\min_{w \in \mathcal{W}} \quad \lambda \|w\|_2^2 + \sum_{(x,y) \in T} \ell(\langle w, f_x \rangle, y)$.
3. Update $\theta_i^{(t+1)}$ according to (an estimate of) its validation loss $\sum_{(x,y) \in V} \ell(\langle w^{(t)}, \phi_i(x) \rangle, y)$.

Observe the equivalence to probabilistic NAS: at each step the classifier (shared parameter) is updated using random feature maps (architectures) on the training samples. The distribution over them is then updated using estimated validation performance. We consider two schemes for this update of $\theta^{(t)}$: (1) exponentiated gradient using the score-function estimate and (2) successive elimination, where we remove a fraction of the feature maps that perform poorly on validation and reassign their probability among the remainder. (1) may be viewed as a softer version of (2), with halving also having only one hyperparameter (elimination rate) and not two (learning rate, stopping criterion).

The first problem we consider is kernel ridge regression over random Fourier features (Rahimi and Recht, 2008) on CIFAR-10. We consider three configuration decisions: data preprocessing, choice of kernel, and bandwidth parameter. This problem was considered by Li et al. (2018), except they fixed the Gaussian kernel whereas we also consider Laplacian; however, they also select the regularization parameter $\lambda$, which weight-sharing does not handle. We also study logistic regression for

| Architecture | Source | Test Error | | Params (M) | Search Cost (GPU Days) | Comparable Search Space? | Search Method |
|---|---|---|---|---|---|---|---|
| | | Best | Average | | | | |
| Shake-Shake | (Devries and Taylor, 2017) | N/A | 2.56 | 26.2 | - | - | manual |
| PyramidNet | (Yamada et al., 2018) | 2.31 | N/A | 26 | - | - | manual |
| NASNet-A* | (Zoph et al., 2018) | N/A | 2.65 | 3.3 | 2000 | N | RL |
| AmoebaNet-B* | (Real et al., 2018) | N/A | $2.55 \pm 0.05$ | 2.8 | 3150 | N | evolution |
| Random search WS† | (Li and Talwalkar, 2019) | 2.71 | $2.85 \pm 0.08$ | 3.8 | 0.7 | Y | random |
| ProxylessNAS | (Cai et al., 2019) | 2.08 | N/A | 5.7 | 4 | N | gradient-based |
| ENAS | (Pham et al., 2018) | 2.89 | N/A | 4.6 | 0.5 | Y | RL |
| ASNG-NAS | (Akimoto et al., 2019) | N/A | $2.83 \pm 0.14$ | 3.9 | 0.1 | Y | gradient-based |
| SNAS | (Xie et al., 2019) | N/A | $2.85 \pm 0.02$ | 2.8 | 1.5 | Y | gradient-based |
| DARTS (first order)† | (Liu et al., 2019) | N/A | $3.00 \pm 0.14$ | 3.3 | 0.4 | Y | gradient-based |
| DARTS (second order)† | (Liu et al., 2019) | N/A | $2.76 \pm 0.09$ | 3.3 | 1 | Y | gradient-based |
| PDARTS | (Chen et al., 2019) | 2.50 | N/A | 3.4 | 0.3 | Y | gradient-based |
| XNAS# | Ours | 2.54 | 2.70 | 3.8 | 0.3 | Y | gradient-based |
| EDARTS† | Ours | 2.54 | $2.69 \pm 0.10$ | 3.9 | 0.4 | Y | gradient-based |

Table 1: **CIFAR-10 Benchmark: Comparison with manual networks and SOTA NAS on final (Stage 3) results.** The results are grouped by those for manually designed networks, full evaluation NAS, weight-sharing NAS, and methods that we evaluated. All test errors are for models trained with auxiliary towers and cutout (parameter counts exclude auxiliary weights). Test errors we report are averaged over 10 seeds. "-" indicates that the field does not apply while "N/A" indicates unknown. Note that search cost is hardware-dependent and our results were procured using Tesla V100 GPUs.
\* We show results for networks with comparable number of parameters.
† For fair comparison to other work, we show the stage 1 cost for a single search trial instead of the four conducted in stage 1. We exclude the cost for stage 2 and 3 evaluation for the same reason.
# We evaluate the reported architecture found by XNAS using the DARTS training scheme for fair comparison.

IMDB sentiment analysis of Bag-of-n-Gram (BonG) featurizations, a standard NLP baseline (Wang and Manning, 2012). Here there are eight configuration decisions: tokenization method, whether to remove stopwords, whether to lowercase, choice of n, whether to binarize features, type of feature weighting, smoothing parameter, and post-processing. As some choices affect the feature dimension we hash the BonGs into a fixed number of bins (Weinberger et al., 2009).

To test the performance of weight-sharing for feature map selection, we randomly sample 64 configurations each for CIFAR-10 and IMDB and examine whether the above schemes converge to the optimal choice. The main comparison method here is thus random search, which runs a full sweep over these samples; by contrast successive halving will need to solve $6 = \log_2 64$ regression problems, while for exponentiated gradient we perform early stopping after five iterations. Note that weight-sharing can do *no better* than random search in terms of accuracy because they are picking a configuration from a space that random search sweeps over. The goal is to see if it consistently returns a good configuration much faster. As our results in Figures 1 and 2 show, successive halving indeed does almost as well as random search in much less time. While exponentiated gradient usually does not recover a near-optimal solution, it does on average return a configuration in the top 10%. We also note the strong benefit of over-parameterization for IMDB – the n-gram vocabulary has size 4 million so the number of bins on the right is much larger than needed to learn in a single-configuration setting. Overall, these experiments show that weight-sharing can also be used as a fast way to obtain signal in regular learning algorithm configuration and not just NAS.

**NAS on CIFAR-10:** Recall from Section 3 that when the architecture space consists of $|E|$ simplices of dimension $|O|$, the convergence rate of exponentiated gradient descent to a stationary point of the objective function is independent of the dimension of the space, while SGD has linear dependence. This result motivates our geometry-aware method called **E**xponentiated-**DARTS** (EDARTS).

EDARTS modifies first-order DARTS in two ways. First, in lieu of the softmax operation used by DARTS on the architecture weights, we use standard normalization so that the weight of operation $o$ on edge $(i, j)$ is $u_o^{(i,j)} = c_o^{(i,j)} / \sum_{o' \in O} c_{o'}^{(i,j)}$. Second, in lieu of Adam, we use exponentiated gradient to update the architecture weights: $c_t = c_{t-1} \exp(-\eta \nabla_c \ell_V (h_{w_{t-1}}(c_{t-1})))$. While EDARTS resembles XNAS (Nayman et al., 2019), our justification for using exponentiated gradient comes directly from aligning with the optimization geometry of ERM. Additionally, EDARTS only requires two straightforward modifications of first-order DARTS, while XNAS relies on a wipeout subroutine and granular gradient-clipping for each edge operation on the cell and data instance level. [1]

---

[1] Our own XNAS implementation informed by correspondence with the authors did not produce competitive results. We still compare to the architecture XNAS reported evaluated by the DARTS training routine in Table 1.

We evaluate EDARTS on the task of designing a CNN cell for CIFAR-10. We use the standard search space as introduced in DARTS (Liu et al., 2019) for evaluation we use the same three stage process used by DARTS and random search with weight-sharing (Li and Talwalkar, 2019), with stage 3 results considered the 'final' results. We provide additional experimental details in Appendix D.

Table 1 shows the performance of EDARTS relative to both manually designed and NAS-discovered architectures. EDARTS finds an architecture that achieves competitive performance with manually designed architectures which have nearly an order-of-magnitude more parameters. Additionally, not only does EDARTS achieve significantly lower test error than first-order DARTS, it also outperforms second order DARTS while requiring less compute time, showcasing the benefit of geometry-aware optimization. Finally, EDARTS achieve comparable performance to the reported architecture for state-of-the-art method XNAS when evaluated using the stage 3 training routine of DARTS.

Following XNAS (Nayman et al., 2019), we also perform an extended evaluation of the best architecture found by EDARTS with AutoAugment, cosine power annealing (Hundt et al., 2019), cross-entropy with label smoothing (Szegedy et al., 2015), and trains for 1500 epochs. We evaluated the XNAS architecture using our implementation for a direct comparison and also to serve as a reproducibility check. EDARTS achieved a test error of $2.18\%$ in the extended evaluation compared to $2.15\%$ for XNAS in our reproduced evaluation; note the published test error for XNAS is $1.81\%$.

To meet a higher bar for reproducibility we report 'broad reproducibility' results by repeating the entire pipeline from stage 1 to stage 3 for two additional sets of seeds. Our results in Table 2 (see Appendix) show that EDARTS has lower variance across experiments than random search with weight sharing (Li and Talwalkar, 2019). However, we do observe non-negligible variance in the performance of the architecture found by different random seed initializations of the shared-weights network, necessitating running multiple searches before selecting an architecture.

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

# A  OPTIMIZATION

This section contains proofs and generalizations of the non-convex optimization results in Section 3.

## A.1  PRELIMINARIES

We first gather some necessary definitions and results from convex analysis.

**Definition A.1.** *Let $\mathcal{X}$ be a convex subset of a finite-dimensional real vector space and $f : \mathcal{X} \mapsto \mathbb{R}$ be everywhere sub-differentiable.*

1. *For $\alpha > 0$, $f$ is $\alpha$-**strongly-convex** w.r.t. norm $\| \cdot \|$ if $\forall\, x, y \in \mathcal{X}$ we have*

$$f(y) \geq f(x) + \langle \nabla f(x), y - x \rangle + \frac{\alpha}{2} \|y - x\|^2$$

2. *For $\beta > 0$, $f$ is $\beta$-**strongly-smooth** w.r.t. norm $\| \cdot \|$ if $\forall\, x, y \in \mathcal{X}$ we have*

$$f(y) \leq f(x) + \langle \nabla f(x), y - x \rangle + \frac{\beta}{2} \|y - x\|^2$$

**Definition A.2.** *Let $\mathcal{X}$ be a convex subset of a finite-dimensional real vector space. The **Bregman divergence** induced by a strictly convex **distance-generating function (DGF)** $\omega : \mathcal{X} \mapsto \mathbb{R}$ is*

$$D_\omega(x||y) = \omega(x) - \omega(y) - \langle \nabla \omega(y), x - y \rangle \,\forall\, x, y \in \mathcal{X}$$

*By definition, the Bregman divergence satisfies the following properties:*

1. *$D_\omega(x||y) \geq 0 \,\forall\, x, y \in \mathcal{X}$ and $D_\omega(x||y) = 0 \iff x = y$.*

2. *If $\omega$ is $\alpha$-strongly-convex w.r.t. norm $\| \cdot \|$ then so is $D_\omega(\cdot||y) \,\forall\, y \in \mathcal{X}$. Furthermore, $D_\omega(x||y) \geq \frac{\alpha}{2} \|x - y\|^2 \,\forall\, x, y \in \mathcal{X}$.*

3. *If $\omega$ is $\beta$-strongly-smooth w.r.t. norm $\| \cdot \|$ then so is $D_\omega(\cdot||y) \,\forall\, y \in \mathcal{X}$. Furthermore, $D_\omega(x||y) \leq \frac{\beta}{2} \|x - y\|^2 \,\forall\, x, y \in \mathcal{X}$.*

**Lemma A.1** (Three-Points Lemma). *(Beck, 2017, Lemma 9.11) For any DGF $\omega : \mathcal{X} \mapsto \mathbb{R}$ and all $x, y, z \in \mathcal{X}$ we have*

$$\langle \nabla \omega(y) - \nabla \omega(x), z - x \rangle = D_\omega(z||x) + D_\omega(x||y) - D_\omega(z||y)$$

**Definition A.3.** *Let $\omega : \mathcal{X} \mapsto \mathbb{R}$ be a 1-strongly-convex DGF. Then for constant $\lambda > 0$ and an everywhere sub-differentiable function $f : \mathcal{X} \mapsto \mathbb{R}$ the **proximal operator** is defined over $x \in \mathcal{X}$ as*

$$\mathrm{prox}_\lambda(x) = \underset{u \in \mathcal{X}}{\arg\min}\, \lambda f(u) + D_\omega(u||x)$$

*Note that the prox operator is well-defined whenever $f$ is $\beta$-strongly-smooth for some $\beta < \lambda$. We will also use the following notation for the **proximal gradient operator**:*

$$\mathrm{prox}\nabla_\lambda(x) = \underset{u \in \mathcal{X}}{\arg\min}\, \lambda \langle \nabla f(x), u \rangle + D_\omega(u||x)$$

*Note that the prox grad operator is always well-defined.*

**Theorem A.1.** *(Beck, 2017, Theorem 9.12) For any $\lambda > 0$, 1-strongly-convex DGF $\omega : \mathcal{X} \mapsto \mathbb{R}$, and $x \in \mathcal{X}$ let $f : \mathcal{X} \mapsto \mathbb{R}$ be an everywhere sub-differentiable function s.t. $\lambda f(\cdot) + D_\omega(\cdot||x)$ is convex over $\mathcal{X}$. Then for $x^+ = \mathrm{prox}_\lambda(x)$ and all $u \in \mathcal{X}$ we have*

$$\langle \nabla \omega(x) - \nabla \omega(x^+), u - x^+ \rangle \leq \lambda(f(u) - f(x^+))$$

**Lemma A.2.** *For any $\lambda > 0$, 1-strongly-convex DGF $\omega : \mathcal{X} \mapsto \mathbb{R}$, and $x \in \mathcal{X}$ let $f : \mathcal{X} \mapsto \mathbb{R}$ be an everywhere sub-differentiable function s.t. $\lambda f(\cdot) + D_\omega(\cdot||x)$ is convex over $\mathcal{X}$. Then*

$$\lambda(f(x) - f(x^+)) \geq D_\omega(x^+||x) + D_\omega(x||x^+)$$

*Proof.* Applying Theorem A.1 followed by Lemma A.1 yields

$$\lambda(f(x) - f(x^+)) \geq \langle \nabla \omega(x) - \nabla \omega(x^+), x - x^+ \rangle = D_\omega(x||x^+) + D_\omega(x^+||x) - D_\omega(x||x)$$

$\square$

**Corollary A.1.** *For any $\lambda > 0$, 1-strongly-convex DGF $\omega : \mathcal{X} \mapsto \mathbb{R}$, $x \in \mathcal{X}$, and everywhere sub-differentiable function $f : \mathcal{X} \mapsto \mathbb{R}$ we have for $x^+ = \mathrm{prox}\nabla_\lambda(x)$ that*

$$\lambda \langle \nabla f(x), x - x^+ \rangle \geq D_\omega(x^+||x) + D_\omega(x||x^+)$$

## A.2 STATIONARITY MEASURES

Because we consider constrained non-convex optimization, we cannot measure convergence to a stationary point by the norm of the gradient. Instead, we analyze the convergence proximal-mapping-based stationarity measures. The most well-known measure is the norm of the *projected gradient* (Ghadimi and Lan, 2013), which in the unconstrained Euclidean case reduces to the norm of the gradient and in the general case measure the distance between the current point and a cone satisfying first-order optimality conditions (Dang and Lan, 2015, Proposition 4.1). Our convergence results hold for a stronger measure that we call the *projected stationarity* and which is inspired by the *Bregman stationarity measure* of Zhang and He (2018) but using the prox grad operator instead of the prox operator.

**Definition A.4.** *Let $\omega : \mathcal{X} \mapsto \mathbb{R}$ be a 1-strongly-convex DGF and $f : \mathcal{X} \mapsto \mathbb{R}$ be an everywhere sub-differentiable function. Then for any $\lambda > 0$ we define the following two quantities:*

1. *The **projected gradient** of $f$ at $x \in \mathcal{X}$ is $\mathcal{G}_\lambda(x) = \frac{x - \mathrm{prox}\nabla_\lambda(x)}{\lambda}$.*

2. *The **projected stationarity** of $f$ at $x \in \mathcal{X}$ is $\Delta_\lambda(x) = \frac{D_\omega(x || \mathrm{prox}\nabla_\lambda(x)) + D_\omega(\mathrm{prox}\nabla_\lambda(x) || x)}{\lambda^2}$.*

*The following properties follow:*

1. *If $\mathcal{X} = \mathbb{R}^d$ for some $d \geq 1$ and $\omega(\cdot) = \frac{1}{2} \| \cdot \|_2^2$ then we have $\mathcal{G}_\lambda(x) = \nabla f(x)$.*

2. *$\|\mathcal{G}_\lambda(x)\|^2 \leq \Delta_\lambda(x) \, \forall \, x \in \mathcal{X}$.*

We can also consider the Bregman stationarity measure of Zhang and He (2018) directly. As this measure depends on the prox operator, which is not always defined, we first state the notion of non-convexity that Zhang and He (2018) consider.

**Definition A.5.** *An everywhere sub-differentiable function $f : \mathcal{X} \mapsto \mathbb{R}$ is $(\gamma, \omega)$-relatively-weakly-convex ($(\gamma, \omega)$-RWC) for $\gamma > 0$ and $\omega : \mathcal{X} \mapsto \mathbb{R}$ a 1-strongly-convex DGF if $f(\cdot) + \gamma\omega(\cdot)$ is convex over $\mathcal{X}$. Note that all $\gamma$-strongly-smooth functions are $(\gamma, \omega)$-RWC.*

Note that $(\gamma, \omega)$-RWC is a generalization of $\gamma$-strong-smoothness w.r.t. the norm w.r.t. which $\omega$ is strongly-convex. Furthermore, for such functions we can always define the prox operator for $\lambda > \gamma$, allowing us to also define the *Bregman gradient* below. Similarly to before, bounding the Bregman stationarity measure yields a stronger notion of convergence than the squared norm of the Bregman gradient. For the relationship between the Bregman stationarity measure and first-order optimality conditions see Zhang and He (2018, Equation 2.11).

**Definition A.6.** *Let $\omega : \mathcal{X} \mapsto \mathbb{R}$ be a 1-strongly-convex DGF and $f : \mathcal{X} \mapsto \mathbb{R}$ be a $(\gamma, \omega)$-RWC everywhere sub-differentiable function for some $\gamma > 0$. Then for any $\lambda > \gamma$ we define the following two quantities:*

1. *The **Bregman gradient** of $f$ at $x \in \mathcal{X}$ is $\mathcal{B}_\lambda(x) = \frac{x - \mathrm{prox}_\lambda(x)}{\lambda}$.*

2. *The **Bregman stationarity** of $f$ at $x \in \mathcal{X}$ is $\Gamma_\lambda(x) = \frac{D_\omega(x || \mathrm{prox}_\lambda(x)) + D_\omega(\mathrm{prox}_\lambda(x) || x)}{\lambda^2}$.*

*Note that, similarly to Definition A.4, $\|\mathcal{B}_\lambda(x)\|^2 \leq \Gamma_\lambda(x) \, \forall \, x \in \mathcal{X}$.*

A.3 SUCCESSIVE STOCHASTIC STRONGLY CONVEX APPROXIMATION FOR CONSTRAINED NON-CONVEX NON-EUCLIDEAN OPTIMIZATION

Here we prove our main optimization results. We begin with a descent lemma guaranteeing improvement of a non-convex function due to approximately optimizing a strongly convex surrogate.

**Lemma A.3.** *Let $\omega : \mathcal{X} \mapsto \mathbb{R}$ be a 1-strongly-convex DGF and $f : \mathcal{X} \mapsto \mathbb{R}$ be everywhere sub-differentiable. For some $x \in \mathcal{X}$ and $\rho > 0$ define $\hat{f}_x(\cdot) = f(\cdot) + \rho D_\omega(\cdot||x)$ and let $\hat{x} \in \mathcal{X}$ be a point s.t. $\mathbb{E}\hat{f}_x(\hat{x}) - \min_{u \in \mathcal{X}} \hat{f}_x(u) \leq \varepsilon$. Then*

1. *If $f$ is $\beta$-strongly-smooth, $\rho > \beta$, and $\lambda = \frac{1}{2\rho}$ then $\frac{\lambda}{2}\Delta_\lambda(x) \leq \mathbb{E}(f(x) - f(\hat{x})) + \varepsilon$.*

2. *If $f$ is $(\gamma, \omega)$-RWC, $\rho > \gamma$, and $\lambda = \frac{1}{2\rho}$ we have that $\frac{\lambda}{2}\Gamma_\lambda(x) \leq \mathbb{E}(f(x) - f(\hat{x})) + \varepsilon$.*

*Proof.* Generalizing an argument in Agarwal et al. (2019, Theorem A.2), for $x^+ \in \mathcal{X}$ we have by strong-convexity of $\hat{f}_x$ that

$$\mathbb{E}(f(x) - f(\hat{x})) \geq \mathbb{E}(f(x) - \hat{f}_x(\hat{x})) = \mathbb{E}\left(f(x) - \min_{u \in \mathcal{X}} \hat{f}(u) + \min_{u \in \mathcal{X}} \hat{f}(u) - \hat{f}(\hat{x})\right)$$
$$\geq f(x) - \hat{f}(x^+) - \varepsilon$$
$$= f(x) - f(x^+) - \rho D_\omega(x^+||x) - \varepsilon$$

If $f$ is $\beta$-strongly-smooth set $x^+ = \text{prox}\nabla_\lambda(x)$, so that by Corollary A.1 we have

$$f(x) - f(x^+) - \rho D_\omega(x^+||x) \geq \langle \nabla f(x), x - x^+ \rangle - \frac{\beta}{2}||x - x^+||^2 - \rho D_\omega(x^+||x)$$
$$\geq \frac{D_\omega(x^+||x) + D_\omega(x||x^+)}{\lambda} - \frac{\rho}{2}||x - x^+||^2 - \rho D_\omega(x^+||x)$$
$$\geq \left(\frac{1}{\lambda} - \rho\right)(D_\omega(x^+||x) + D_\omega(x||x^+))$$
$$= \frac{\lambda}{2}\Delta_\lambda(x)$$

In the other case of $f$ being $(\gamma, \omega)$-RWC set $x^+ = \text{prox}_\lambda(x)$, so that by Lemma A.2 we have

$$f(x) - f(x^+) - \rho D_\omega(x^+||x) \geq \frac{D_\omega(x^+||x) + D_\omega(x||x^+)}{\lambda} - \rho D_\omega(x^+||x)$$
$$\geq \rho(D_\omega(x^+||x) + D_\omega(x||x^+))$$
$$= \frac{\lambda}{2}\Gamma_\lambda(x)$$

$\square$

We now turn to formalizing our multi-block setting and assumptions.

**Setting A.1.** *For $i = 1, \ldots, b$ let $\mathcal{X}_i$ be a convex subset of a real vector space with an associated DGF $\omega_i : \mathcal{X}_i \mapsto \mathbb{R}$ that is 1-strongly-convex w.r.t. some norm $\|\cdot\|_{(i)}$ over $\mathcal{X}_i$. We have an everywhere sub-differentiable function $f : \mathcal{X} \mapsto \mathbb{R}$ over the product space $\mathcal{X} = \times_{i=1}^{b} \mathcal{X}_i$.*

Our main results will hold for the case when the following general assumption is satisfied. We will later show how this assumption can follow from strong smoothness or relative weak convexity and existing algorithmic results.

**Assumption A.1.** *In Setting A.1, for any given $\varepsilon > 0$ and each $i \in [b]$ there exists a constant $\rho_i > 0$ and an algorithm $\mathcal{A}_i : \mathcal{X} \mapsto \mathcal{X}$ that takes a point $x \in \mathcal{X}$ and returns a point $\hat{x} \in \mathcal{X}$ satisfying $\hat{x}_{-i} = x_{-i}$ and*

$$\mathbb{E}_{\mathcal{A}_i}(f(\hat{x}_i, x_{-i}) + \rho_i D_{\omega_i}(\hat{x}_i||x_i)) \leq \varepsilon + \min_{u \in \mathcal{X}_i} f(u, x_{-i}) + \rho_i D_{\omega_i}(u||x_i)$$

*where the subscript $i$ selects block $i$, the subscript $-i$ selects all blocks other than block $i$, and $\mathbb{E}_{\mathcal{A}_i}$ denotes expectation w.r.t. the randomness of algorithm $\mathcal{A}_i$ and any associated stochastic oracles.*

---

**Algorithm 2:** Generic successive convex approximation algorithm for reaching a stationary point of the non-convex function in Setting A.1.

---

**Input:** Point $x^{(1)} \in \mathcal{X}$ in the product space of Setting A.1. Algorithms $\mathcal{A}_1, \ldots, \mathcal{A}_b : \mathcal{X} \mapsto \mathcal{X}$.

**for** iteration $t \in [T]$ **do**

    sample $\xi_t \sim \text{Unif}[b]$

    update $x^{(t+1)} \leftarrow \mathcal{A}_{\xi_t}(x^{(t)})$

**Output:** $x \sim \text{Unif}\{x^{(t)}\}_{t=1}^{T}$.

---

Our main result relies on the following simple lemma guaranteeing non-convex convergence for a generic measure satisfying guaranteed expected descent:

**Lemma A.4.** *In Setting A.1, for some $\varepsilon > 0$ and each $i \in [b]$ let $E_{\lambda_i}^{(i)}(x)$ be any measure s.t. for some $\lambda_i$ and some algorithm $\mathcal{A}_i : \mathcal{X} \mapsto \mathcal{X}$ we have $\frac{\lambda_i}{2} E_{\lambda_i}^{(i)} \leq \mathbb{E}_{\mathcal{A}_i}(f(x) - f(\mathcal{A}_i(x))) + \varepsilon \; \forall \, x \in \mathcal{X}$. Then the output $x$ of Algorithm 2 satisfies*

$$\mathbb{E} \sum_{i=1}^{b} \lambda_i E_{\lambda_i}^{(i)}(x) \leq 2b \left( \frac{F}{T} + \varepsilon \right)$$

*where $F = f(x^{(1)}) - \arg\min_{u \in \mathcal{X}} f(x)$ and the expectation is taken over the sampling at each iteration, the sampling of the output, and the randomness of the algorithms and any associated stochastic oracles.*

*Proof.* Define $\Xi_t = \{(\xi_s, \mathcal{A}_{\xi_s})\}_{s=1}^{t}$ and note that $x^{(t+1)} = \mathcal{A}_{\xi_t}(x^{(t)})$. We then have

$$F = f(x^{(1)}) - \arg\min_{u \in \mathcal{X}} f(x) \geq \mathbb{E}_{\Xi_T}(f(x^{(1)}) - f(x^{(T+1)}))$$

$$= \mathbb{E}_{\Xi_T} \sum_{t=1}^{T}(f(x^{(t)}) - f(x^{(t+1)}))$$

$$= \sum_{t=1}^{T} \mathbb{E}_{\Xi_{t-1}} \mathbb{E}_{\xi_t} \mathbb{E}_{\mathcal{A}_{\xi_t}}(f(x^{(t)}) - f(x^{(t+1)}))$$

$$= \frac{1}{b} \sum_{t=1}^{T} \mathbb{E}_{\Xi_{t-1}} \sum_{i=1}^{b} \mathbb{E}_{\mathcal{A}_i}(f(x^{(t)}) - f(x^{(t+1)}))$$

$$\geq \frac{1}{b} \mathbb{E}_{\Xi_T} \sum_{t=1}^{T} \sum_{i=1}^{b} \frac{\lambda_i}{2} E_{\lambda_i}^{(i)}(x^{(t)}) - \varepsilon$$

which implies that

$$\mathbb{E} \sum_{i=1}^{b} \lambda_i E_{\lambda_i}^{(i)}(x) = \frac{1}{T} \mathbb{E}_{\Xi_T} \sum_{t=1}^{T} \sum_{i=1}^{b} \lambda_i E_{\lambda_i}^{(i)}(x^{(t)}) \leq 2b \left( \frac{F}{T} + \varepsilon \right)$$

$\square$

In the single-block setting, Lemmas A.3 and A.4 directly imply the following guarantee:

**Theorem A.2.** *In Setting A.1 and under Assumption A.1, let $b = 1$ and $\rho$ satisfy one of the following:*

1. $f : \mathcal{X} \mapsto \mathbb{R}$ *is $\beta$-strongly-smooth and $\rho = 2\beta$.*

2. $f : \mathcal{X} \mapsto \mathbb{R}$ *is $(\gamma, \omega)$-RWC for some DGF $\omega : \mathcal{X} \mapsto \mathbb{R}$ and $\rho = 2\gamma$.*

*Then Algorithm 2 returns a point $x \in \mathcal{X}$ satisfying one of the following (respectively, w.r.t. the above settings) for $F = f(x^{(1)}) - \min_{u \in \mathcal{X}} f(u)$*

    *1.* $\mathbb{E}\Delta_{\frac{1}{4\beta}}(x) \leq 8\beta \left( \frac{F}{T} + \varepsilon \right)$          *2.* $\mathbb{E}\Gamma_{\frac{1}{4\gamma}}(x) \leq 8\gamma \left( \frac{F}{T} + \varepsilon \right)$

*Here the expectation is taken over the randomness of the algorithm and oracle.*

We can apply a known result for the strongly convex case to recover the rate of Zhang and He (2018) for non-convex stochastic mirror descent, up to an additional depending on $\omega$:

**Corollary A.2.** *In Setting A.1 for $b = 1$ and $(\gamma, \omega)$-RWC $f$, suppose we have access to $f$ through a stochastic gradient oracle $g(x) = \mathbb{E}\nabla f(x)$ such that $\mathbb{E}\|g\|_*^2 \leq G^2$. Let $\mathcal{A} : \mathcal{X} \mapsto \mathcal{X}$ be an algorithm that for any $x \in \mathcal{X}$ runs the Epoch-GD method of Hazan and Kale (2014) with total number of steps $N$, initial epoch length $T_1 = 4$ and initial learning rate $\eta_1 = \frac{1}{\gamma}$ on $\hat{f}_x(\cdot) = f(\cdot) + 2\gamma D_\omega(\cdot\|x)$. Then with $NT$ calls to the stochastic gradient oracle Algorithm 2 returns a point $x \in \mathcal{X}$ satisfying*

$$\mathbb{E}\Gamma_{\frac{1}{4\gamma}}(x) \leq 8\gamma \left( \frac{F}{T} + \frac{16(G^2 + 4\gamma^2 L_\omega^2)}{\gamma N} \right)$$

*for $F = f(x^{(1)}) - \min_{u \in \mathcal{X}} f(x)$ and $L_\omega$ the Lipschitz constant of $\omega$ w.r.t. $\|\cdot\|$ over $\mathcal{X}$. So an expected $\varepsilon$-stationary-point, as measured by $\mathbb{E}\sqrt{\Gamma_{\frac{1}{4\gamma}}}$, can be reached in $\mathcal{O}\left( \frac{\gamma F(G^2 + \gamma^2 L_\omega^2)}{\varepsilon^4} \right)$ oracle calls.*

*Proof.* Apply Theorem 5 of Hazan and Kale (2014) together with the fact that $\hat{f}_x$ is $\gamma$-strongly-convex w.r.t. $\|\cdot\|$ and its stochastic gradient is bounded by $G^2 + 4\gamma^2 L_\omega$. □

For the multi-block case our results hold only the projected stationarity measure:

**Theorem A.3.** *In Setting A.1 and under Assumption A.1 assume $f(\cdot, x_{-i})$ is $\beta$-strongly-smooth w.r.t. $\|\cdot\|_{(i)}$ over $\mathcal{X}_i$ for each $i \in [b]$ and each $x_{-i} \in \mathcal{X}_{-i}$. If $\rho = \rho_1 = \cdots \rho_b = 2\beta$ and $F = f(x^{(1)}) - \min_{u \in \mathcal{X}} f(u)$ then Algorithm 2 returns a point $x \in \mathcal{X}$ satisfying*

$$\mathbb{E}\Delta_{\frac{1}{4\beta}}(x) \leq 8\beta b \left( \frac{F}{T} + \varepsilon \right)$$

*Here the expectation is taken over the randomness of the algorithm and oracle and the projected stationarity measure $\Delta_\lambda$ is defined w.r.t. the Bregman divergence of the DGF $\omega(x) = \sum_{i=1}^b \omega_i(x_i)$.*

*Proof.* For each block $i \in [b]$ the conditions of Lemma A.4 are satisfied by assumption and applying Lemma A.3 to the projected stationary measures

$$\Delta_\lambda^{(i)}(x) = \frac{D_{\omega_i}(x_i\|\operatorname{prox}\nabla_\lambda^{(i)}(x)) + D_{\omega_i}(\operatorname{prox}\nabla_\lambda^{(i)}(x)\|x_i))}{\lambda^2}$$

where $\operatorname{prox}\nabla_\lambda^{(i)}(x) = \arg\min_{u \in \mathcal{X}_i} \lambda\langle \nabla_i f(x), u\rangle + D_{\omega_i}(u\|x_i)$. To apply Lemma A.4 in the multi-block setting it suffices to show that the sum of the projected stationarity measures on each block is equal to the projected stationarity measure induced by the sum of the DGFs. For some $\lambda > 0$ and any $i \in [b]$ we have that

$$\begin{aligned}
\operatorname{prox}\nabla_\lambda(x)_i &= \left( \arg\min_{u \in \mathcal{X}} \lambda\langle\nabla f(x), u\rangle + D_\omega(u\|x) \right)_i \\
&= \left( \arg\min_{u \in \mathcal{X}} \lambda \sum_{i=1}^b \langle\nabla_i f(x), u_i\rangle + D_{\omega_i}(u_i\|x_i) \right)_i \\
&= \arg\min_{u \in \mathcal{X}_i} \lambda\langle\nabla_i f(x), u\rangle + D_{\omega_i}(u\|x_i) = \operatorname{prox}\nabla_\lambda^{(i)}(x)
\end{aligned}$$

and so

$$\begin{aligned}
\Delta_\lambda(x) &= \frac{D_\omega(x\|\operatorname{prox}\nabla_\lambda(x)) + D_\omega(\operatorname{prox}\nabla_\lambda(x)\|x)}{\lambda^2} \\
&= \frac{1}{\lambda^2} \sum_{i=1}^b D_{\omega_i}(x_i\|\operatorname{prox}\nabla_\lambda(x)_i) + D_{\omega_i}(\operatorname{prox}\nabla_\lambda(x)_i\|x_i) \\
&= \frac{1}{\lambda^2} \sum_{i=1}^b D_{\omega_i}(x_i\|\operatorname{prox}\nabla_\lambda^{(i)}(x)) + D_{\omega_i}(\operatorname{prox}\nabla_\lambda^{(i)}(x)\|x_i) = \sum_{i=1}^b \Delta_\lambda^{(i)}(x)
\end{aligned}$$

Thus applying Lemma A.2 with $\lambda = \frac{1}{4\rho}$ yields the result. □

In the following corollary we recover the rate of Dang and Lan (2015) for non-convex block-stochastic mirror descent, up to an additional term depending on $\omega_i$:

**Corollary A.3.** *In Setting A.1 for $\beta$-strongly-smooth $f$, suppose we have access to $f$ through a stochastic gradient oracle $g(x) = \mathbb{E}\nabla f(x)$ such that $\mathbb{E}\|g_i\|_{(i),*}^2 \leq G_i^2$. For $i \in [b]$ let $\mathcal{A}_i : \mathcal{X} \mapsto \mathcal{X}$ be an algorithm that for any $x \in \mathcal{X}$ runs the Epoch-GD method of Hazan and Kale (2014) with total number of steps $N$, initial epoch length $T_1 = 4$ and initial learning rate $\eta_1 = \frac{1}{\gamma}$ on surrogate function $\hat{f}_x(\cdot) = f(\cdot, x_{-i}) + 2\beta D_{\omega_i}(\cdot\|x_i)$. Then with $NT$ calls to the stochastic gradient oracle Algorithm 2 returns a point $x \in \mathcal{X}$ satisfying*

$$\mathbb{E}\Delta_{\frac{1}{4\beta}}(x) \leq 8\beta b \left( \frac{F}{T} + \frac{16}{\beta N} \max_{i \in [b]} G_i^2 + 4\beta^2 L_{\omega_i}^2 \right)$$

*for $F = f(x^{(1)}) - \min_{u \in \mathcal{X}} f(x)$ and $L_{\omega_i}$ the Lipschitz constant of $\omega_i$ w.r.t. $\|\cdot\|_{(i)}$ over $\mathcal{X}_i$. So an expected $\varepsilon$-stationary-point, as measured by $\mathbb{E}\sqrt{\Delta_{\frac{1}{4\beta}}}$, can be reached in a number of stochastic oracle calls bounded by*

$$\mathcal{O}\left( \frac{\beta F b^2}{\varepsilon^4} \sum_{i=1}^{b} G_i^2 + \beta^2 L_{\omega_i}^2 \right)$$

We can specialize this to the architecture search setting where we have a configuration search space contained in the product of simplices induced by having $n$ decisions with $c$ choices each together with a parameter space bounded in Euclidean norm.

**Corollary A.4.** *Under the assumptions of Corollary A.3, suppose $b = 2$ and we have the following two geometries:*

- $\mathcal{X}_1 = \bigtimes_{i=1}^{n} \{p \in [\delta, 1]^c : \|p\|_1 = 1\}$, $\|\cdot\|_{(1)} = \|\cdot\|_1$, $\omega_1(\cdot) = \langle \cdot, \log(\cdot) \rangle$.

- $\mathcal{X}_2 = \{w \in \mathbb{R}^d : \|w\|_2 \leq B\}$, $\|\cdot\|_{(2)} = \|\cdot\|_2$, $\omega_2(\cdot) = \frac{1}{2}\|\cdot\|_2^2$.

*Suppose the stochastic gradient oracle of $f$ has bounded $\ell_\infty$-norm $\sigma_1$ over $\mathcal{X}_1$ and $\sigma_2$ over $\mathcal{X}_2$. Then Algorithm 2 will return an expected $\varepsilon$-stationary point of $f$ under the projected stationarity measure in a number of stochastic oracle calls bounded by*

$$\mathcal{O}\left( \frac{\beta F b^2}{\varepsilon^4} \left( \sigma_1^2 + \beta^2 \log \frac{1}{\delta} + \sigma_2^2 d + \beta^2 B^2 \right) \right)$$

# B    GENERALIZATION

This section contains proofs of the generalization results in Section 4.

## B.1    SETTINGS AND MAIN ASSUMPTION

We first describe the setting for which we prove our general result.

**Setting B.1.** *Let $\mathcal{C}$ be a set of possible architecture/configurations of finite size such that each $c \in \mathcal{C}$ is associated with a parameterized hypothesis class $H_c = \{h_w^{(c)} : X \mapsto Y : w \in \mathcal{W}\}$ for input space $Z = X \times Y$ and fixed set of possible weights $\mathcal{W}$. We will measure the performance of a hypothesis $h_w^{(c)}$ on an input $z = (x, y) \in Z$ using $\ell_z(w, c) = \ell(h_w^{(c)}(x), y)$ for some B-bounded loss $\ell : Y \times Y \mapsto [-B, B]$.*

*We are given a training sample $T \sim \mathcal{D}^{m_T}$ of size $m_T$ and a validation sample $V \sim \mathcal{D}^{m_V}$ of size $m_V$, where $\mathcal{D}$ is some distribution over the input space $Z$. Abusing notation, we will denote the the population risk by $\ell_{\mathcal{D}}(h_w^{(c)}) = \ell_{\mathcal{D}}(w, c) = \mathbb{E}_{z \sim \mathcal{D}} \ell_z(w, c)$ and for any finite subset $S \subset Z$ we will denote the empirical risk by $\ell_S(h_w^{(c)}) = \ell_S(w, c) = \frac{1}{|S|} \sum_{z \in S} \ell_z(w, c)$.*

*Finally, we will consider solutions of optimization problems that depend on the training data and architecture. Specifically, for any configuration $c \in \mathcal{C}$ and finite subset $S \subset Z$ let $\mathcal{W}_c(S) \subset \mathcal{W}$ be the set of global minima of some optimization problem induced by $S$ and $c$ and let the associated version space (Kearns et al., 1997) be $H_c(S) = \{h_w^{(c)} : w \in \mathcal{W}_c(S)\}$.*

We next give as examples two specific settings encompassed by Setting B.1.

**Setting B.2.** *For **feature map selection**, in Setting B.1 the configuration space $\mathcal{C}$ is a set of feature maps $f_c : X \mapsto \mathbb{R}^d$, the set of weights $\mathcal{W} \subset \mathbb{R}^d$ consists of linear classifiers, for inputs $x \in X$ the hypotheses are $h_w^{(c)}(x) = \langle f_c(x), w \rangle$ for $w \in \mathcal{W}$, and so $\mathcal{W}_c(S)$ is the singleton set of solutions to the regularized ERM problem*

$$\underset{w \in \mathcal{W}}{\arg\min} \quad \lambda \|w\|_2^2 \quad + \sum_{(x,y) \in S} \ell(\langle f_c(x), w \rangle, y)$$

*for some coefficient $\lambda > 0$.*

**Setting B.3.** *For **neural architecture search**, in Setting B.1 the configuration space consists of all possible choices of edges on a DAG of $N$ nodes and a choice from one of $K$ operations at each edge, for a total number of configurations bounded by $K^{N^2}$. The hypothesis $h_w^{(c)} : X \mapsto Y$ is determined by a choice of architecture $c \in \mathcal{C}$ and a set of network weights $w \in \mathcal{W}$ and the loss $\ell : Y \times Y \mapsto \{0, 1\}$ is the zero-one loss. In the simplest case $\mathcal{W}_c(S)$ is the set of global minima of the ERM problem*

$$\min_{w \in \mathcal{W}} \quad \sum_{(x,y) \in S} \ell(h_w^{(c)}(x), y)$$

We now state the main assumption we require.

**Assumption B.1.** *In Setting B.1 there exists a good architecture $c^* \in \mathcal{C}$, i.e. one satisfying $(w^*, c^*) \in \arg\min_{\mathcal{W} \times \mathcal{C}} \ell_{\mathcal{D}}(w, c)$ for some weights $w^* \in \mathcal{W}$, such that w.p. $1 - \delta$ over the drawing of training set $T \sim \mathcal{D}^{m_T}$ at least one of the minima of the optimization problem induced by $c^*$ and $T$ has low excess risk, i.e. $\exists w \in \mathcal{W}_{c^*}(T)$ s.t.*

$$\ell_{\mathcal{D}}(w, c^*) - \ell_{\mathcal{D}}(w^*, c^*) \leq \varepsilon_{c^*}(m_T, \delta)$$

*for some error function $\varepsilon_{c^*}$.*

Clearly, we prefer error functions $\varepsilon_{c^*}$ that are decreasing in the number of training samples $m_T$ and increasing at most poly-logarithmically in $\frac{1}{\delta}$. This assumption requires that if we knew the optimal configuration *a priori*, then the provided optimization problem will find a good set of weights for it. We will show how, under reasonable assumptions, Assumption B.1 can be formally shown to hold in Settings B.2 and B.3.

## B.2 MAIN RESULT

Our general result will be stated in terms of covering numbers of certain function classes.

**Definition B.1.** *Let $H$ be a class of functions from $X$ to $Y$. For any $\varepsilon > 0$ the associated $L_\infty$* **covering number** $N(H, \varepsilon)$ *of $H$ is the minimal positive integer $k$ such that $H$ can be covered by $k$ balls of $L^\infty$-radius $\varepsilon$.*

The following is then a standard result in statistical learning theory (see e.g. Lafferty et al. (2010, Theorem 7.82)):

**Theorem B.1.** *Let $H$ be a class of functions from $X$ to $Y$ and let $\ell : Y \times Y \mapsto [0, B]$ be an $L$-Lipschitz, $B$-bounded loss function. Then for any distribution $\mathcal{D}$ over $X \times Y$ we have*

$$\Pr_{S \sim \mathcal{D}^m} \left( \sup_{h \in H} |\ell_\mathcal{D}(h) - \ell_S(h)| \geq 3\varepsilon \right) \leq 2N(H, \varepsilon) \exp \left( -\frac{m\varepsilon^2}{2B^2} \right)$$

*where we use the loss notation from Setting B.1.*

Before stating our theorem, we define a final quantity, which measures the log covering number of the version spaces induced by the optimization procedure over a given training set.

**Definition B.2.** *In Setting B.1, for any sample $S \subset X \times Y$ define the **version entropy** to be $\Lambda(H, \varepsilon, S) = \log N \left( \bigcup_{c \in \mathcal{C}} H_c(S), \varepsilon \right)$.*

**Theorem B.2.** *In Setting B.1 let $(\hat{w}, \hat{c}) \in \mathcal{W} \times \mathcal{C}$ be obtained as a solution to the following optimization problem:*

$$\arg\min_{w \in \mathcal{W}, c \in \mathcal{C}} \ell_V(w, c) \qquad \text{s.t.} \qquad w \in \mathcal{W}_c(T)$$

*Then under Assumption B.1 we have w.p. $1 - 3\delta$ that*

$$\ell_\mathcal{D}(\hat{w}, \hat{c}) \leq \ell_\mathcal{D}(w^*, c^*)$$
$$+ \varepsilon_{c^*}(m_T, \delta) + B\sqrt{\frac{1}{2m_V} \log \frac{1}{\delta}} + \inf_{\varepsilon > 0} 3\varepsilon + B\sqrt{\frac{2}{m_V} \left( \Lambda(H, \varepsilon, T) + \log \frac{1}{\delta} \right)}$$

*Proof.* We have that

$$\ell_\mathcal{D}(\hat{w}, \hat{c}) - \ell_\mathcal{D}(w^*, c^*) \quad \leq \quad \underbrace{\ell_\mathcal{D}(\hat{w}, \hat{c}) - \ell_V(\hat{w}, \hat{c})}_{1} \quad + \quad \underbrace{\ell_V(\hat{w}, \hat{c}) - \ell_V(w, c^*)}_{2}$$
$$+ \quad \underbrace{\ell_V(w, c^*) - \ell_\mathcal{D}(w, c^*)}_{3} \quad + \quad \underbrace{\ell_\mathcal{D}(w, c^*) - \ell_\mathcal{D}(w^*, c^*)}_{4}$$

each term of which can be bounded as follows:

1. Since $\hat{w} \in \mathcal{W}_{\hat{c}}(T)$ for some $\hat{c} \in \mathcal{C}$ the hypothesis space can be covered by the union of the coverings of $H_c(T)$ over $c \in \mathcal{C}$, so by Theorem B.1 we have that w.p. $1 - \delta$

$$\ell_\mathcal{D}(\hat{w}, \hat{c}) - \ell_V(\hat{w}, \hat{c}) \leq \inf_{\varepsilon > 0} 3\varepsilon + B\sqrt{\frac{2}{m_V} \left( \Lambda(H, \varepsilon, T) + \log \frac{1}{\delta} \right)}$$

2. By optimality of the pair $(\hat{w}, \hat{c})$ and the fact that $w \in \mathcal{W}_{c^*}(T)$ we have

$$\ell_V(\hat{w}, \hat{c}) = \min_{c \in \mathcal{C}, w \in \mathcal{W}_c(T)} \ell_V(\hat{w}, \hat{c}) \leq \min_{w \in \mathcal{W}_{c^*}(T)} \ell_V(w, c^*) \leq \ell_V(w, c^*)$$

3. Hoeffding's inequality yields $\ell_V(w, c^*) - \ell_\mathcal{D}(w, c^*) \leq B\sqrt{\frac{1}{2m_V} \log \frac{1}{\delta}}$ w.p. $1 - \delta$

4. Assumption B.1 states that $\ell_\mathcal{D}(w, c^*) - \ell_\mathcal{D}(w^*, c^*) \leq \varepsilon_{c^*}(m_T, \delta)$ w.p. $1 - \delta$.

$\square$

### B.3 APPLICATIONS

For the feature map selection problem, Assumption B.1 holds by standard results for $\ell_2$-regularized ERM for linear classification (e.g. Sridharan et al. (2008)):

**Corollary B.1.** *In Setting B.2, suppose the loss function $\ell$ is Lipschitz. Then for regularization parameter $\lambda = \sqrt{\frac{1}{m_T} \log \frac{1}{\delta}}$ we have*

$$\ell_{\mathcal{D}}(w, c^*) - \ell_{\mathcal{D}}(w^*, c^*) \leq \mathcal{O}\left( \sqrt{\frac{\|w^*\|_2^2 + 1}{m_T} \log \frac{1}{\delta}} \right)$$

We can then directly apply Theorem B.2 and the fact that the version entropy is bounded by $\log |\mathcal{C}|$ because the minimizer over the training set is always unique to get the following:

**Corollary B.2.** *In Setting B.2 let $(\hat{w}, \hat{c}) \in \mathcal{W} \times \mathcal{C}$ be obtained as a solution to the following optimization problem:*

$$\underset{w \in \mathcal{W}, c \in \mathcal{C}}{\arg\min} \quad \ell_V(w, c) \qquad s.t. \qquad w = \underset{w \in \mathcal{W}}{\arg\min} \quad \lambda \|w\|_2^2 + \sum_{(x,y) \in T} \ell(\langle f_c(x), w \rangle, y)$$

*Then*

$$\ell_{\mathcal{D}}(\hat{w}, \hat{c}) - \ell_{\mathcal{D}}(w^*, c^*) \leq \mathcal{O}\left( \sqrt{\frac{\|w^*\|_2^2 + 1}{m_T} \log \frac{1}{\delta}} + \sqrt{\frac{1}{m_V} \log \frac{|C| + 1}{\delta}} \right)$$

In the special case of kernel selection we can apply generalization results for learning with random features to show that we can compete with the optimal RKHS from among those associated with one of the configurations (Rudi and Rosasco, 2017, Theorem 1):

**Corollary B.3.** *In Setting B.2, suppose each configuration $c \in \mathcal{C}$ is associated with a random Fourier feature approximation of a continuous shift-invariant kernel that approximates an RKHS $\mathcal{H}_c$. Suppose $\ell$ is the squared loss so that $(\hat{w}, \hat{c}) \in \mathcal{W} \times \mathcal{C}$ is obtained as a solution to the following optimization problem:*

$$\underset{w \in \mathcal{W}, c \in \mathcal{C}}{\arg\min} \quad \ell_V(w, c) \qquad s.t. \qquad w = \underset{w \in \mathcal{W}}{\arg\min} \quad \lambda \|w\|_2^2 + \sum_{(x,y) \in T} (\langle f_c(x), w \rangle - y)^2$$

*If the number of random features $d = \mathcal{O}(\sqrt{m_T} \log \sqrt{m_T}/\delta)$ and $\lambda = 1/\sqrt{m_T}$ then w.p. $1 - \delta$ we have*

$$\ell_{\mathcal{D}}(h_{\hat{w}}^{\hat{c}}) - \min_{c \in \mathcal{C}} \min_{h \in \mathcal{H}_c} \ell_{\mathcal{D}}(h) \leq \mathcal{O}\left( \frac{\log^2 \frac{1}{\delta}}{\sqrt{m_T}} + \sqrt{\frac{1}{m_V} \log \frac{|C| + 1}{\delta}} \right)$$

In the case of neural architecture search we are often solving (unregularized) ERM in the inner optimization problem. In this case we can make an assumption weaker than Assumption B.1, namely that the set of empirical risk minimizers contains a solution that, rather than having low excess risk, simply has low generalization error; then applying Hoeffding's inequality yields the following:

**Corollary B.4.** *In Setting B.1 let $(\hat{w}, \hat{c}) \in \mathcal{W} \times \mathcal{C}$ be obtained as a solution to the following optimization problem:*

$$\underset{w \in \mathcal{W}, c \in \mathcal{C}}{\arg\min} \quad \ell_V(w, c) \qquad s.t. \qquad w \in \underset{w' \in \mathcal{W}}{\arg\min} \ell_T(w', c)$$

*Suppose there exists $c^* \in \mathcal{C}$ satisfying $(w^*, c^*) \in \arg\min_{\mathcal{W} \times \mathcal{C}} \ell_{\mathcal{D}}(w, c)$ for some weights $w^* \in \mathcal{W}$ such that w.p. $1 - \delta$ over the drawing of training set $T \sim \mathcal{D}^{m_T}$ at least one of the minima of the optimization problem induced by $c^*$ and $T$ has low generalization error, i.e. $\exists\, w \in \arg\min_{w' \in \mathcal{W}} \ell_T(w', c^*)$ s.t.*

$$\ell_{\mathcal{D}}(w, c^*) - \ell_T(w^*, c^*) \leq \varepsilon_{c^*}(m_T, \delta)$$

*for some error function $\varepsilon_{c^*}$. Then we have w.p. $1 - 4\delta$ that*

$$\ell_{\mathcal{D}}(\hat{w}, \hat{c}) \leq \ell_{\mathcal{D}}(w^*, c^*) + \varepsilon_{c^*}(m_T, \delta) + B\sqrt{\frac{1}{2m_V} \log \frac{1}{\delta}} + B\sqrt{\frac{1}{2m_T} \log \frac{1}{\delta}}$$

$$+ \inf_{\varepsilon > 0} 3\varepsilon + B\sqrt{\frac{2}{m_V} \left( \Lambda(H, \varepsilon, T) + \log \frac{1}{\delta} \right)}$$

## C  FEATURE MAP SELECTION DETAILS

Solvers for Ridge regression and logistic regression were from scikit-learn (Pedregosa et al., 2011). For CIFAR-10 we use the kernel configuration setting from Li et al. (2018) but replacing the regularization parameter by the option to use the Laplace kernel instead of Gaussian. The regularization was fixed to $\lambda = \frac{1}{2}$ The split is 40K/10K/10K. For IMDB we consider the following configuration choices:

1. Tokenizer: removing punctuation, splitting on punctuation, custom NLTK (Loper and Bird, 2002)

2. removing stopwords or not

3. lowercasing or not

4. order: $n = 1, 2, 3$, higher order includes all lower-order $n$-grams

5. binarizing features or not

6. feature weights: naive-Bayes (Wang and Manning, 2012) or smoothed inverse frequency (Arora et al., 2017)

7. $\alpha$: smoothing parameter for this weighting

8. preprocessing: None, $\ell_2$-normalization, averaging by number of tokens

The regularization was fixed to $C = 1$. The split is 25K/12.5K/12.5K

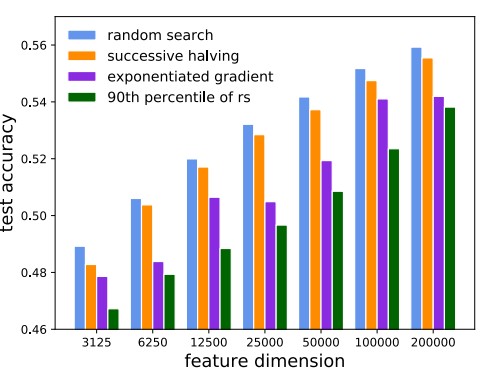 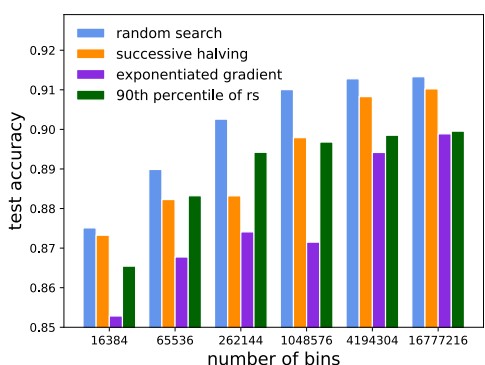

Figure 3: Validation accuracy on CIFAR-10 (left) and IMDB (right) of feature map selection with weight-sharing compared to a full sweep of random configurations. Average over 16 seeds.

## D  NAS EXPERIMENT DETAILS

We provide additional details on CNN NAS benchmark on designing CNN cells for CIFAR-10 below.

### D.1  SEARCH SPACE

We consider the same search space as DARTS (Liu et al., 2019), which has become one of the standard search spaces for CNN cell search (Nayman et al., 2019; Chen et al., 2019; Xie et al., 2019; Noy et al., 2019; Liang et al., 2019). Following DARTS and random search with weight-sharing (Li and Talwalkar, 2019), our evaluation of EDARTS consists of three stages:

- **Stage 1:** In the search phase, we run EDARTS with four random seeds to reduce variance from different initialization of the shared-weights network.
- **Stage 2:** We evaluate the best architecture identified by each search run by training from scratch.
- **Stage 3:** We perform a more thorough evaluation of the best architecture from stage 2 by training with ten different random seed initializations.

For stages 2 and 3, we train each architecture for 600 epochs with the same hyperparameter settings as DARTS.

For completeness, we describe the convolutional neural network search space considered. The set of operations $O$ considered at each node include: (1) $3 \times 3$ separable convolution, (2) $5 \times 5$ separable convolution, (3) $3 \times 3$ dilated convolution, (4) $5 \times 5$ dilated convolution, (5) max pooling, (6) average pooling, (7) identity.

We use the same search space to design a "normal" cell and a "reduction" cell; the normal cells have stride 1 operations that do not change the dimension of the input, while the reduction cells have stride 2 operations that half the length and width dimensions of the input. In the experiments, for both cell types, we set $N = 6$ with 2 input nodes and 4 intermediate nodes, after which the output of all intermediate nodes are concatenated to form the output of the cell.

### D.2 STAGE 1: ARCHITECTURE SEARCH

We use EDARTS to train a smaller shared-weights network in the search phase with 8 layers and 24 initial channels instead of the 16 used by DARTS. Additionally, to more closely mirror the architecture used for evaluation in stage 2 and 3, we use an auxiliary head with weight 0.4 and scheduled path dropout of 0.2. For the EDARTS architecture updates, we use a learning rate of 0.2 for the normal cell and 0.6 for the reduction cell. All other hyperparameters are the same as DARTS: 50 training epochs, batch size of 64, gradient clipping of 5 for network weights, SGD with momentum set to 0.9 and learning rate annealed from 0.025 to 0.001 with cosine annealing (Loshchilov and Hutter, 2016), and weight decay of 0.0003.

### D.3 STAGE 2 AND 3: ARCHITECTURE EVALUATION

We use the same evaluation scheme as DARTS when retraining architectures from scratch. The larger evaluation network has 20 layers and 36 initial channels and is trained for 600 epochs using SGD with momentum set to 0.9, a batch size of 96, and a learning rate of 0.025 annealed down to 0; the gradient clipping scheduled drop path rate and weight decay are identical to the search phase. We also use an auxiliary head (Szegedy et al., 2015) with a weight of 0.4 and cutout (Devries and Taylor, 2017).

### D.4 DOES WEIGHT-SHARING PERFORM IMPLICIT REGULARIZATION

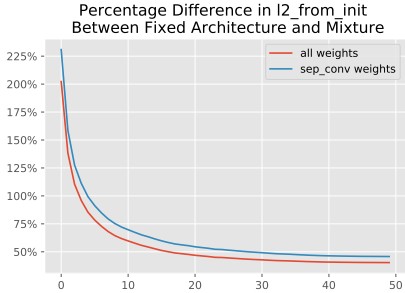 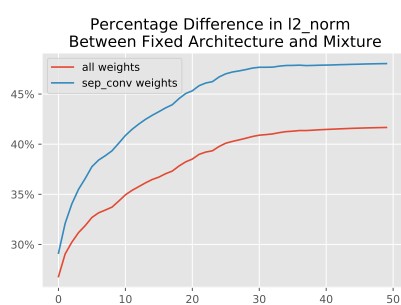

Figure 4: Norm comparison between weight of the best EDARTS architecture trained from scratch versus that of the corresponding shared-weights network.

We investigate whether weight-sharing implicitly regularizes the hypothesis space by examining the $\ell_2$ norms and distance from initialization of the shared-weights network relative to that observed when training the best EDARTS architecture from scratch. We use the same network depth and hyperparameters as those used for the shared-weights network to train the fixed architecture. Figure 4 shows the percent difference in the norms between the fixed architecture and the shared-weights network pruned to just the operations kept for the fixed architecture. From the chart, we can see that both the $\ell_2$ distance from initialization and the $\ell_2$ norm of the shared-weights is smaller than that of a fixed network are higher than that of the shared-weights network by over $40\%$, suggesting weight-sharing acts as a form of implicit regularization.

| Set 1 | | | | | Set 2 | | | | | Set 3 | | | | |
|---|---|---|---|---|---|---|---|---|---|---|---|---|---|---|
| 1 | 2 | 3 | 4 | Stage 3 | 1 | 2 | 3 | 4 | Stage 3 | 1 | 2 | 3 | 4 | Stage 3 |
| 2.79 | 2.91 | 2.99 | 2.88 | 2.79 | 3.55 | 3.20 | 3.05 | 2.65 | **2.69** | 2.74 | 2.96 | 2.63 | 2.61 | 2.71 |

Table 2: EDARTS stage 2 and 3 test error for 3 sets of random seeds. The bolded test error is the best stage 3 performance across the 3 sets of random seeds.

## D.5 DISCUSSION OF REPRODUCIBILITY IN NAS

Our 'broad reproducibility' results in Table 2 show that EDARTS has lower variance across experiments than random search with weight sharing (Li and Talwalkar, 2019). However, we do observe non-negligible variance in the performance of the architecture found by different random seed initializations of the shared-weights network, necessitating running multiple searches before selecting an architecture.

