# OpenReview forum: "On Weight-Sharing and Bilevel Optimization in Architecture Search"
_ICLR.cc/2020/Conference — Reject_

### Official Review · AnonReviewer1 · 2019-10-18
**Official Blind Review #1**

**Rating:** 3

**Review:**

This work proposes an algorithm for handling the weight-sharing neural architecture search problem. It also derives generalization bound for this problem.

The reviewer has several concerns:

1) the SBMD and ASCA algorithms are existing generic algorithms. The analysis in this work also looks very generic. There is a sense of disconnection with the considered training problems. The reviewer would like to see more discussions on how to connect the algorithms with specific NAS problems. For example, what is the beta parameter when training a NAS problem?

2) The convergence rate improvement brought by using mirror descent has been long known. It is not easy to see what is the contribution of this work.

3) The generalization part seems to be meaningful. But it may be much stronger if the NAS problem can also have a theoretical bound. It is less appealing to only discuss cases with strongly convex objectives.

**Experience Assessment:**

I do not know much about this area.

**Review Assessment: Checking Correctness Of Derivations And Theory:**

I did not assess the derivations or theory.

**Review Assessment: Checking Correctness Of Experiments:**

I did not assess the experiments.

**Review Assessment: Thoroughness In Paper Reading:**

I made a quick assessment of this paper.

---

> ### Author Response · Authors · 2019-11-15
> **Response 1**
>
> Response: Thank you for your comments. We hope to address your issues below:
>
> 1) Novelty and relevance of SBMD and ASCA to NAS:
> - Novelty: We respectfully disagree with your comment.  In fact, our work is the first to introduce ASCA and it is *not* an existing generic algorithm.
> - Beta parameter: The beta parameter depends on the activation functions used and on the data. As we acknowledged at submission, this restricts the cases where the theory applies to smooth activation functions (sigmoid, tanh).
>
> 2) Contribution of work on top of existing results for mirror descent:
> - While mirror descent is indeed a well-known approach in the optimization literature, its connection to NAS has not been explored. Our theoretical guarantees are largely motivated by this connection and provide significant improvements over existing analysis for NAS (Akimoto et al., 2019; Carlucci et al., 2019; Nayman et al., 2019; Noy et al., 2019; Yao et al., 2019).
> - The guarantees we provide for the ASCA variant of mirror descent are *new* and not previously known in any form outside the Euclidean case.
>
> 3) Generalization bounds for NAS
> - We *do* provide a theoretical bound for NAS.  The main generalization result (Theorem 4.1) can be applied to non-convex inner objectives, including for NAS. We discuss what the result means for NAS starting at the bottom of page 7, with reference to existing theoretical work on complexity of the set of local minima of deep nets and a discussion of what further understanding can be gained.

---

### Official Review · AnonReviewer2 · 2019-10-22
**Official Blind Review #2**

**Rating:** 3

**Review:**

I have not worked in the optimization filed and I am only gently followed the NAS field. I might under-valued the theoretical contribution.

This work provides  theoretical analysis for the NAS using weight sharing in two aspects:
1) The authors give non-asymptotic stationary-point convergence guarantees (based on stochastic block mirror descent (SBMD) from Dang and Lan (2015)) for the empirical risk minimization (ERM) objective associated with weight-sharing. Based on this analysis, the authors proposed to use  exponentiated gradient to update architecture parameter, which enjoys faster convergence rate than the original results in Dang and Lan (2015). The author also provided an alternative to SBMD that uses alternating successive convex approximation (ASCA) which has similar convergence rate.
2) The author provide generalization guarantees for this objective over structured hypothesis spaces associated with a finite set of architectures.

My biggest concern is the validity of the proposed exponentiated gradient update, at least empirically. We indeed observed slightly improvement in test error over DARTS on the CIFAR10 benchmark but how reproducible the results are? Can you compare at least on the other benchmark (PENN TREEBANK) used in Liu et al 2019? Also, comparing to first order DARTS, search cost is the same and this is hard to justify the better convergence rate for EDARTS. In addition, the results on feature map selection is not very encouraging as the gap to the successive halving is significant.

The author proposed ASCA, as an alternative method to SBMD. Why we need such alternative? What is the advantage of ASCA comparing to SBMD? When should I use ASCA and when SBMD? How do they empirically different?

Then I feel some wording can be improved. For example, "while requiring computation training …”,  “…which may be of independent interest”.



**Experience Assessment:**

I do not know much about this area.

**Review Assessment: Checking Correctness Of Derivations And Theory:**

I did not assess the derivations or theory.

**Review Assessment: Checking Correctness Of Experiments:**

I assessed the sensibility of the experiments.

**Review Assessment: Thoroughness In Paper Reading:**

I read the paper at least twice and used my best judgement in assessing the paper.

---

> ### Author Response · Authors · 2019-11-15
> **Response 2**
>
> Response: Thank you for your comments. We hope to address your issues below:
>
> Theoretical analysis:
> We would like to emphasize that the convergence guarantees improve significantly upon several previous NAS analyses (Akimoto et al., 2019; Carlucci et al., 2019; Nayman et al., 2019; Noy et al., 2019; Yao et al., 2019). To our knowledge they are the first results that are both non-asymptotic (finite-time convergence) and optimize a quantity of direct interest (empirical risk objective).
>
> Validity of exponentiated-gradient update:
> (1) NAS experiments: While we agree that the experiments would benefit from an additional dataset, we decided to focus on CIFAR-10 due to the high computational cost associated with running these experiments.  Similar to Li & Talwalkar 2019, we have also observed that the variance associated with stage 3 evaluation of architectures is much higher on the Penn Treebank dataset and chose to instead focus our resources in thoroughly evaluating EDARTS on the lower variance CIFAR-10 benchmark.  As stated in the last paragraph of the paper, we follow a higher bar for reproducibility than many other NAS publications (e.g., DARTS, SNAS, XNAS, ASAP, ProxylessNAS, etc) and report results for EDARTS for 3 different sets of seeds on CIFAR-10; EDARTS reaches ~2.70% test error on 2 out of the 3 runs.  We have not seen similar broad reproducibility results for other NAS methods.
> (2) Same search cost as first-order DARTS: the search cost is the same since we train for the same number of epochs.  The faster convergence rate is reflected in the better resulting architecture.
> (3) Kernel experiments: Please note that the kernel experiments were motivated by understanding weight-sharing and its generalization guarantees on a simpler problem (kernel ridge regression), not as a test of the performance of our optimizer. As a result, the successive halving method that exceeds exponentiated-gradient on those experiments is also an algorithm proposed in this paper, and it may be viewed as a hard-cutoff version of exponentiated-gradient. Furthermore, successive halving would be difficult to apply directly in the larger NAS search space.
>
> ASCA vs. SBMD:
> (1) Need for such an alternative: The motivation behind our paper is to theoretically understand NAS methods. Several NAS methods have found it useful to run many iterations on both the shared-weights and architecture-weights before switching (e.g. ENAS by Pham et al., 2018 and MdeNAS by Zheng et al., 2019). This approach is reflected in the ASCA algorithm and not in SBMD.
> (2) Respective advantages: While most (but not all, as discussed above) NAS methods prefer an SBMD-style approach, ASCA may be preferable when fast solvers are available for strongly-convex relaxations of the problem at hand.
>
> Wording:
> Thank you for pointing these out - they will be corrected.

---

### Decision · Program_Chairs · 2019-12-19

**Decision:**

Reject

**Comment:**

Since there were only two official reviews submitted, I reviewed the paper to form a third viewpoint.  I agree with reviewer 2 on the following points, which support rejection of the paper:
1) Only CIFAR is evaluated without Penn Treebank;
2) The "faster convergence" is not empirically justified by better final accuracy with same amount of search cost; and
3) The advantage of the proposed ACSA over SBMD is not clearly demonstrated in the paper.

The scores of the two official reviews are insufficient for acceptance, and an additional review did not overturn this view.